# α-DPO: Adaptive Reward Margin is What Direct Preference Optimization Needs

## Abstract

Aligning large language models (LLMs) with human values and intentions is crucial for their utility, honesty, and safety. Reinforcement learning from human feedback (RLHF) is a popular approach to achieve this alignment, but it faces challenges in computational efficiency and training stability. Recent methods like Direct Preference Optimization (DPO) and Simple Preference Optimization (SimPO) have proposed offline alternatives to RLHF, simplifying the process by reparameterizing the reward function. However, DPO depends on a potentially suboptimal reference model, and SimPO's assumption of a fixed target reward margin may lead to suboptimal decisions in diverse data settings. In this work, we propose α-DPO, an adaptive preference optimization algorithm designed to address these limitations by introducing a dynamic reward margin. Specifically, α-DPO employs an adaptive preference distribution, balancing the policy model and the reference model to achieve personalized reward margins. We provide theoretical guarantees for α-DPO, demonstrating its effectiveness as a surrogate optimization objective and its ability to balance alignment and diversity through KL divergence control. Empirical evaluations on AlpacaEval 2 and Arena-Hard show that α-DPO consistently outperforms DPO and SimPO across various model settings, establishing it as a robust approach for fine-tuning LLMs. Our method achieves significant improvements in win rates, highlighting its potential as a powerful tool for LLM alignment. The code is available at
https://anonymous.4open.science/r/alpha-DPO-E0D3.

## 1 Introduction

Learning from human feedback is essential for aligning large language models (LLMs) with human values and intentions (Leike et al., 2018), ensuring they are helpful, honest, and harmless (Askell et al., 2021). Reinforcement learning from human feedback (RLHF) (Christiano et al., 2017; Ouyang et al., 2022; Stiennon et al., 2020) is a widely used method for fine-tuning LLMs to achieve this goal. However, RLHF faces challenges, particularly in computational efficiency and training stability due to its multi-stage process. Recently, alternative offline algorithms like DPO (Rafailov et al., 2023) and SimPO (Meng et al., 2024) have been explored. Specifically, DPO reparameterizes the reward function in RLHF to directly learn a policy model ($\pi_\theta$) from preference data, removing the need for an explicit reward model. Building on DPO, SimPO further simplifies the process by eliminating the need for a reference model ($\pi_{\text{ref}}$), and introduces a target reward margin $\gamma$ to enlarge the distance between the response pair, thereby achieving strong performance. This naturally raises the question:

*Do we really need a reference model in the alignment process?*

Motivated by this question, we examine SimPO: it can be viewed as a variant of DPO where the original reference model $\pi_{\text{ref}}$ is effectively replaced by an *implicit reference model* $\hat{\pi}_{\text{ref}}$. In SimPO, the target reward margin $\gamma$ actually reflects a constant difference between the log likelihoods of a selected response and a rejected one, *i.e.,* $(\log \hat{\pi}_{\text{ref}}(y_w|x) - \log \hat{\pi}_{\text{ref}}(y_l|x))$. As the constant difference $\gamma$ is independent of arbitrary responses, this implicitly assumes a uniform reference distribution (*cf.* Figure 1). By tuning $\gamma$, SimPO effectively finds an "ideal" uniform *implicit reference model*, leading to substantial performance improvements over standard DPO, particularly when the original reference model $\pi_{\text{ref}}$ is suboptimal (Hong et al., 2024).

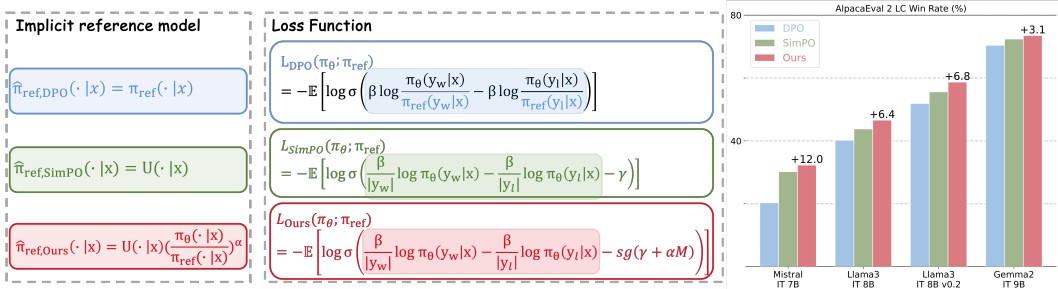

Figure 1: DPO, SimPO and $\alpha$-DPO mainly differ in their *implicit reference model*, as indicated in the shaded box, leading to variations in their respective loss functions. $\alpha$-DPO outperforms DPO and SimPO across a wide range of settings on AlpacaEval 2.

While conceptually appealing with empirical improvements, SimPO has two inherent limitations: (1) Applying the same target reward margin to all pairwise comparisons ignores the variability in the data (Yang et al., 2024; Wu et al., 2024), potentially leading to suboptimal decisions in some cases; and (2) The implicit assumption of a uniform reference model somehow lacks a solid theoretical foundation. These limitations could hinder the model's ability to achieve alignment across varied training data, especially in domains with diverse preferences or complex reward structures (Morimura et al., 2024; He et al., 2024).

To address these limitations of SimPO, we propose an adaptive preference distribution, which leads to an adaptive reward margin for different response pairs. We term this simple yet effective preference optimization algorithm $\alpha$-DPO. Specifically, the adaptive preference distribution is heuristically set as: $\hat{\pi}_{\text{ref}}(y|x) = U(y|x)\left(\pi_\theta(y|x)/\pi_{\text{ref}}(y|x)\right)^\alpha$. Here, $U(y|x)$, inspired by SimPO, employs a uniform distribution to establish an initial target reward margin, while the term $\left(\pi_\theta(y|x)/\pi_{\text{ref}}(y|x)\right)^\alpha$ adjusts the balance between the policy model $\pi_\theta$ and the reference model $\pi_{\text{ref}}$ to achieve a personalized reward margin. When $\alpha = 0$, $\alpha$-DPO reduces to SimPO; as $\alpha$ increases, the ratio between $\pi_\theta$ and $\pi_{\text{ref}}$ becomes dominant, enabling a personalized, dynamic target. More important, $\alpha$-DPO offers several intriguing theoretical insights:

- **Theoretical Guarantee via Lower Bound**: We prove that $\alpha$-DPO's objective serves as a lower bound on the online SimPO loss. This theoretical guarantee justifies the use of $\alpha$-DPO as a surrogate optimization objective and ensures that optimizing it leads to improved policy performance and generalization.

- **Balancing Alignment and Diversity**: We demonstrate that $\alpha$-DPO balances alignment and diversity via KL divergence control. By approximating the sequential KL divergence between the policy and the reference model, $\alpha$-DPO achieves computational efficiency and robustness, particularly when the reference model is not well-calibrated at the token level.

Extensive analysis indicates that $\alpha$-DPO leverages preference data more effectively by assigning personalized margins to each pair, resulting in an improved policy model. As demonstrated in Figure 1, our method consistently outperforms DPO and SimPO across three base model settings (Mistral2-7B, Llama3-8B, and Gemma2-9B) on AlpacaEval 2 and Arena-Hard (*cf.* Section 5). Notably, we achieve a 58.7 length-controlled win rate on AlpacaEval 2, and a 35.7 win rate on Arena-Hard, establishing it as the strongest 8B open-source model to date.

## 2 PRELIMINARIES

**Offline Alignment.** In the offline alignment problem, we have access to a dataset $\mathcal{D} = \{(x, y_w, y_l)\}$ comprising prompts $x$ and labeled response pairs $(y_w, y_l)$ obtained from a reference policy $\pi_{\text{ref}}$. Here, $y_w$ is the preferred (winning) response and $y_l$ is the less preferred (losing) response. Although the underlying latent reward function $r^*(x, y)$ that governs these preferences is not directly observable, the Bradley-Terry (BT) model (Bradley & Terry, 1952) provides a framework for modeling pairwise comparisons:

$$\mathbb{P}(y_w \succ y_l|x) = \frac{\exp(r^*(x, y_w))}{\exp(r^*(x, y_w)) + \exp(r^*(x, y_l))}, \tag{1}$$

where $r^*(x, y)$ assigns a latent reward to each response $y$ given prompt $x$. The goal of offline alignment is to learn a policy $\pi_\theta$ that approximates $r^*(x, y)$ using $\mathcal{D}$.

**Reinforcement Learning from Human Feedback (RLHF).** Classical offline alignment algorithms employ reinforcement learning with a KL-regularized reward objective (Bai et al., 2022; Ziegler et al., 2019; Ouyang et al., 2022), defined for a regularization parameter $\eta > 0$:

$$\max_{\pi_\theta} \mathbb{E}_{x \sim \mathcal{D}, y \sim \pi_\theta(y|x)}[r_\phi(x, y)] - \beta \mathbb{D}_{\text{KL}}[\pi_\theta(y|x) || \pi_{\text{ref}}(y|x)], \tag{2}$$

where $r_\phi(x, y)$ is the reward function learned using the BT model on the preference dataset, $\pi_\theta$ is the policy model being optimized, $\pi_{\text{ref}}$ is the fixed reference policy, typically obtained via supervised fine-tuning. The KL-divergence regularizes the policy to remain close to the reference model.

**Directed Preference Optimization (DPO).** DPO (Rafailov et al., 2023) is a leading offline preference optimization method. Instead of learning an explicit reward model, DPO reparameterizes the reward function $r(x, y)$ using a closed-form expression involving the optimal policy:

$$r(x, y) = \beta \log \frac{\pi_\theta(y|x)}{\pi_{\text{ref}}(y|x)} + \beta \log Z(x), \tag{3}$$

where $Z(x)$ is the partition function independent of $y$. This leads to the DPO loss for any triplet $(x, y_w, y_l)$:

$$\mathcal{L}_{\text{DPO}}(\pi_\theta; \pi_{\text{ref}}) = -\mathbb{E}_{(x, y_w, y_l) \sim \mathcal{D}} \left[ \log \sigma \left( \beta \log \frac{\pi_\theta(y_w|x)}{\pi_\theta(y_l|x)} - \beta \log \frac{\pi_{\text{ref}}(y_w|x)}{\pi_{\text{ref}}(y_l|x)} \right) \right]. \tag{4}$$

where $\sigma(\cdot)$ denotes the sigmoid function.

**Simple Preference Optimization (SimPO).** SimPO (Meng et al., 2024) introduces two key contributions: (1) a length-normalized reward, calculated as the average log-probability per token of a response under the policy model $\pi_\theta$, and (2) a target reward margin $\gamma$ to ensure the reward difference between winning and losing responses exceeds this margin. The SimPO loss is formulated as:

$$\mathcal{L}_{\text{SimPO}}(\pi_\theta) = -\mathbb{E}_{(x, y_w, y_l) \sim \mathcal{D}} \left[ \log \sigma \left( \frac{\beta}{|y_w|} \log \pi_\theta(y_w|x) - \frac{\beta}{|y_l|} \log \pi_\theta(y_l|x) - \gamma \right) \right], \tag{5}$$

where $|y|$ denotes the length (*i.e.*, number of tokens) of response $y$, normalizing the reward by response lengths, and $\gamma$ is the target reward margin.

# 3 METHOD

In this section, we establish a unified framework that connects DPO and SimPO (Section 3.1), highlighting the critical role of the reference model in preference optimization. We then introduce $\alpha$-DPO (Section 3.2), a novel preference optimization algorithm that synergizes the strengths of both DPO and SimPO.

## 3.1 A COMMON FRAMEWORK FOR DPO AND SIMPO

A key insight in our work is that SimPO implicitly adopts a uniform distribution over the vocabulary as its reference model, whereas DPO employs the SFT model as the reference. By examining the role of the reference model in both methods, we derive the following result:

**Theorem 3.1.** *Let $U(y|x)$ denote a uniform distribution over the vocabulary for a given input $x$, replacing $\pi_{ref}(y|x)$ in the DPO loss function. Then, the DPO loss function simplifies to:*

$$\mathcal{L}(\pi_\theta; U) = -\mathbb{E}_{(x, y_w, y_l) \sim \mathcal{D}} \left[ \log \sigma \left( \beta \left( \log \pi_\theta(y_w|x) - \log \pi_\theta(y_l|x) \right) - \gamma \right) \right], \tag{6}$$

*where $\gamma = \beta \left( \log U(y_w|x) - \log U(y_l|x) \right)$ is a constant. Under a length-normalized reward formulation, this loss function becomes:*

$$\mathcal{L}_{LN}(\pi_\theta; U) = -\mathbb{E}_{(x, y_w, y_l) \sim \mathcal{D}} \left[ \log \sigma \left( \frac{\beta}{|y_w|} \log \pi_\theta(y_w|x) - \frac{\beta}{|y_l|} \log \pi_\theta(y_l|x) - \gamma \right) \right]. \tag{7}$$

*Therefore, SimPO can be interpreted as a special case of DPO where the reference model is a uniform distribution.*

**Why does uniform policy assign different probabilities to winning and losing responses:** The selection mechanism biases the probability distributions of winning and losing responses, even though both are sampled from the same SFT model. A reward model assigns scores to responses, selecting the highest-scoring as $y_w$ and the lowest-scoring as $y_l$. This process changes the effective corpora for winners and losers, leading to different values for the uniform distribution over each.

Theorem 3.1 establishes a unified framework for DPO and SimPO by showing that replacing the reference model $\pi_{\text{ref}}$ in DPO with a uniform distribution $U$ reduces the DPO loss to the SimPO loss, up to a constant term $\gamma$. This reveals that SimPO is essentially DPO with a uniform reference model. Consequently, the term $\beta \left( \log \pi_{\text{ref}}(y_w|x) - \log \pi_{\text{ref}}(y_l|x) \right)$ collapses to a constant, emphasizing the pivotal role of the implicit reference model in preference optimization.

**Limitations of DPO:** As depicted in Figure 2, the reference model $\pi_{\text{ref}}$ in DPO may not effectively distinguish between the preferred ($y_w$) and less preferred ($y_l$) responses, as its outputs do not inherently reflect the preference information. In contrast, using a uniform distribution as in SimPO results in a reward margin $\gamma$, ensuring that the reward difference between the preferred and less preferred responses is entirely governed by the policy model $\pi_\theta$.

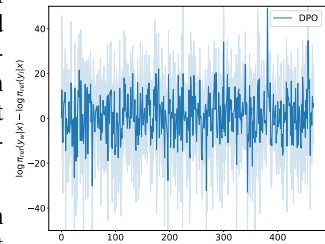

Figure 2: $\log \pi_{\text{ref}}(y_w|x) - \log \pi_{\text{ref}}(y_l|x)$ along the training steps.

**Limitations of SimPO**: While SimPO simplifies the loss function by using a constant offset $\gamma$, it overlooks the variability inherent in different data instances, as $\gamma$ remains the same across all training samples. This rigidity could lead to suboptimal performance, especially in the presence of noise or inconsistencies in the data. Moreover, completely discarding the original reference model $\pi_{\text{ref}}$ may fail to capture important distinctions between response pairs that could be leveraged to improve learning.

### 3.2 PROPOSED METHOD: $\alpha$-DPO

Our analysis highlights the significant impact of the reference model in preference optimization. To overcome the limitations identified in both DPO and SimPO, we propose the following principles:

**Principle 1**: *The reference model should contribute to differentiating between preferred and less preferred responses.*

**Principle 2**: *The reference model should adapt to discrepancies between response pairs to capture instance-specific nuances.*

Principle 1 addresses the shortcoming in DPO, where the reference model may inadequately distinguish between $y_w$ and $y_l$, introducing uncertainty without a guaranteed margin. Principle 2 rectifies the oversimplification in SimPO, where the absence of a reference model fails to account for variability across different instances.

**Deriving the $\alpha$-DPO Objective.** Starting from the standard Reinforcement Learning (RL) objective for preference optimization, we redefine the reference model $\pi_{\text{ref}}$ as an *implicit reference model* $\hat{\pi}_{\text{ref}}$, formulated as:

$$\hat{\pi}_{\text{ref}}(y|x) \propto U(y|x) \left( \pi_\theta(y|x)/\pi_{\text{ref}}(y|x) \right)^\alpha, \tag{8}$$

where $\alpha$ is a hyperparameter controlling the influence of the policy model on the reference model, and $U(y|x)$ is a uniform distribution serving as a constant baseline. When $\alpha = 0$, $\hat{\pi}_{\text{ref}}$ reduces to the uniform distribution as in SimPO; when $\alpha = 1$, it incorporates the ratio between the policy and reference models as in DPO. We provide the motivation for Equation 8 in Appendix B.

Substituting $\hat{\pi}_{\text{ref}}$ into the original DPO loss function, we obtain the $\alpha$-DPO objective:

$$\begin{aligned}
\mathcal{L}_{\alpha\text{-DPO}}(\pi_\theta, \pi_{\text{ref}}) &= -\mathbb{E}_{(x,y_w,y_l)\sim\mathcal{D}} \left[ \log \sigma \left( \beta \log \frac{\pi_\theta(y_w|x)}{\pi_\theta(y_l|x)} - \beta \log \frac{\hat{\pi}_{\text{ref}}(y_w|x)}{\hat{\pi}_{\text{ref}}(y_l|x)} \right) \right] \\
&= -\mathbb{E}_{(x,y_w,y_l)\sim\mathcal{D}} \left[ \log \sigma \left( \beta \left( \log \frac{\pi_\theta(y_w|x)}{\pi_\theta(y_l|x)} \right) - \left[ \gamma + \alpha M(x,y_w,y_l) \right] \right) \right],
\end{aligned} \tag{9}$$

where $\gamma = \beta \left( \log \frac{U(y_w|x)}{U(y_l|x)} \right)$ is a constant offset as before, and $M(x,y_w,y_l)$ is defined as:

$$M(x,y_w,y_l) = \beta \left( \log \frac{\pi_\theta(y_w|x)\pi_{\text{ref}}(y_l|x)}{\pi_{\text{ref}}(y_w|x)\pi_\theta(y_l|x)} \right). \tag{10}$$

The term $M(x, y_w, y_l)$ measures the divergence between the policy model $\pi_\theta$ and the reference model $\pi_{\text{ref}}$ over the response pairs, effectively capturing instance-specific discrepancies as described in Principle 2.

**Stop Gradient on $\hat{\pi}_{\text{ref}}$**: Although $\hat{\pi}_{\text{ref}}$ depends on $\pi_\theta$ and $\pi_{\text{ref}}$, it is intended to serve as a fixed reference during optimization. To prevent gradients from backpropagating through $\hat{\pi}_{\text{ref}}$ to $\pi_\theta$, we apply a stop-gradient operation, denoted as $\text{sg}[\cdot]$, ensuring that $\hat{\pi}_{\text{ref}}$ remains constant during the policy updates.

**Normalization of $M(x, y_w, y_l)$**: To stabilize training and avoid $M(x, y_w, y_l)$ dominating the loss due to scale variations, we apply Z-score normalization (Patro & Sahu, 2015) to M:

$$M^*(x, y_w, y_l) = \frac{M(x, y_w, y_l) - \mu_M}{\sigma_M}, \qquad (11)$$

where $\mu_M$ and $\sigma_M$ are the mean and standard deviation of $M$ computed over the training dataset.

**Length-normalized Reward Formulation**: Inspired by the technique used in SimPO, we incorporate length normalization into our method. This adjustment ensures that rewards are scaled appropriately with respect to the length of the sequences, thereby stabilizing the training process. As demonstrated in our experiments (*cf.* Appendix C.2), we also confirmed that even without length normalization, our method remains effective and continues to show performance improvements.

**Final Objective**: Incorporating the above considerations, the final $\alpha$-DPO loss function becomes:

$$\mathcal{L}_{\alpha\text{-DPO}}(\pi_\theta, \pi_{\text{ref}}) = -\mathbb{E}_{(x,y_w,y_l)\sim\mathcal{D}} \left[ \log \sigma \left( u(x, y_w, y_l) - \text{sg}\left[\gamma + \alpha M^*(x, y_w, y_l)\right] \right) \right], \qquad (12)$$

where $u(x, y_w, y_l) = \frac{\beta}{|y_w|} \log \pi_\theta(y_w|x) - \frac{\beta}{|y_l|} \log \pi_\theta(y_l|x)$. This formulation ensures balanced influence between the policy and reference models, aligning with Principles 1 and 2. By incorporating the normalized discrepancy term $M^*(x, y_w, y_l)$, $\alpha$-DPO adaptively adjusts the margin between preferred and less preferred responses based on instance-specific differences, enhancing learning.

## 4 THEORETICAL ANALYSIS OF $\alpha$-DPO

In this section, we provide a theoretical analysis of $\alpha$-DPO by connecting its objective to the online SimPO loss. We also explore how $\alpha$-DPO manages the trade-off between alignment and diversity through KL divergence control.

### 4.1 RELATION OF $\alpha$-DPO TO ONLINE SIMPO

In this section, we relate the $\alpha$-DPO objective to the online SimPO loss and demonstrate how our preference optimization method leads to a generalization bound for the policy model. We consider the following definitions:

**Definition 4.1** (Online SimPO Loss). Let $\pi_\theta$ represent the current policy and $\pi_{\theta_{\text{old}}}$ represent the policy from a previous iteration. The online SimPO loss is defined as:

$$\mathcal{L}_{\text{SimPO}}^{\text{online}}(\pi_\theta, \pi_{\text{ref}}) = -\mathbb{E}_{(x,y_w,y_l)\sim\pi_{\theta_{\text{old}}}} \left[ \log \sigma \left( \frac{\beta}{|y_w|} \log \pi_\theta(y_w|x) - \frac{\beta}{|y_l|} \log \pi_\theta(y_l|x) - \gamma \right) \right], \quad (13)$$

where $\sigma(\cdot)$ is the sigmoid function, $\beta$ is a scaling parameter, $|y|$ represents the sequence length, and $\gamma$ is a margin parameter.

By applying importance sampling, this loss can be rewritten as:

$$\mathcal{L}_{\text{SimPO}}^{\text{online}}(\pi_\theta, \pi_{\text{ref}}) = -\mathbb{E}_{(x,y_w,y_l)\sim\pi_{\text{ref}}} \left[ w(y_w, y_l|x) \log \sigma \left( \frac{\beta}{|y_w|} \log \pi_\theta(y_w|x) - \frac{\beta}{|y_l|} \log \pi_\theta(y_l|x) - \gamma \right) \right], \qquad (14)$$

where the importance weight $w(y_w, y_l|x)$ is given by:

$$w(y_w, y_l|x) = \frac{\pi_{\theta_{\text{old}}}(y_w|x)}{\pi_{\text{ref}}(y_w|x)} \cdot \frac{\pi_{\theta_{\text{old}}}(y_l|x)}{\pi_{\text{ref}}(y_l|x)}. \qquad (15)$$

However, this importance weighting scheme can lead to undesirable results when $\pi_{\theta_{old}}(y_l|x) > \pi_{ref}(y_l|x)$, particularly for rejected responses. In such cases, the weight assigned to less preferred responses becomes disproportionately large, which is counterintuitive and can negatively impact the optimization process. To address this issue, we propose an alternative weighting scheme:

$$w_{corr}(y_w, y_l|x) = \frac{\pi_{\theta_{old}}(y_w|x)}{\pi_{ref}(y_w|x)} \cdot \frac{\pi_{ref}(y_l|x)}{\pi_{\theta_{old}}(y_l|x)}. \tag{16}$$

This corrected weighting inversely scales the weight of overrepresented rejected responses, effectively reducing their influence in the expectation and improving the alignment between $\pi_\theta$ and $\pi_{ref}$. While this adjustment deviates from the standard importance sampling framework, it functions as a variance reduction technique that biases the estimator towards more desirable properties. Similar approaches have been explored in importance sampling literature to mitigate variance issues (Hanna et al., 2019; Patterson et al., 2021). We now establish that the $\alpha$-DPO objective serves as a lower bound on the online SimPO loss, justifying its use as a surrogate optimization objective.

**Lemma 4.2** (Tight bound between $\alpha$-DPO and online SimPO loss)**.** *For any policy model $\pi_\theta$ and reference model $\pi_{ref}$, there exists a sufficiently small $\alpha > 0$ such that the following inequalities hold:*

$$|\mathcal{L}_{SimPO}^{online}(\pi_\theta, \pi_{ref}) - \mathcal{L}_{\alpha\text{-}DPO}(\pi_\theta, \pi_{ref})| \leq \varepsilon(\alpha),$$

*where*

$$\varepsilon(\alpha) = \mathbb{E}_{\pi_{ref}}\left[\alpha|B| \, |\log \sigma(A) - \sigma(A) + 1|\right],$$

$A = \frac{\beta}{|y_w|}\log \pi_\theta(y_w|x) - \frac{\beta}{|y_l|}\log \pi_\theta(y_l|x) - \gamma, \text{ and } B = \log \frac{\pi_\theta(y_w|x)}{\pi_{ref}(y_w|x)} - \log \frac{\pi_\theta(y_l|x)}{\pi_{ref}(y_l|x)}.$

Lemma 4.2 establishes that our objective $\mathcal{L}_{\alpha\text{-}DPO}(\pi_\theta, \pi_{ref})$ tightly approximates the online SimPO loss $\mathcal{L}_{SimPO}^{online}(\pi_\theta, \pi_{ref})$ within a controllable margin $\varepsilon(\alpha)$ that depends linearly on $\alpha$. By choosing $\alpha$ to be sufficiently small, we can make $\varepsilon(\alpha)$ arbitrarily close to zero, ensuring that the gap between $\mathcal{L}_{\alpha\text{-}DPO}$ and $\mathcal{L}_{SimPO}^{online}$ is negligible.

### 4.2 Balancing Alignment and Diversity via KL Divergence Control

Balancing alignment performance with response diversity is crucial in recent alignment methods (Zeng et al., 2024; Wang et al., 2024a; Ji et al., 2024a). A popular approach is the Token-Level Direct Preference Optimization (TDPO) method (Zeng et al., 2024), which introduces fine-grained control of the KL divergence at the token level. Given a prompt $x$ and preceding tokens $y^{<t}$, the policy $\pi_\theta$ generates the next token $z$ by sampling from $\pi_\theta(z|x, y^{<t})$.

By mapping the reward model to a token-level format, the TDPO loss is defined as:

$$\mathcal{L}_{TDPO}(\pi_\theta) = -\mathbb{E}_{(x,y_w,y_l)\sim\mathcal{D}}\left[\log \sigma\left(\beta \log \frac{\pi_\theta(y_w|x)}{\pi_{ref}(y_w|x)} - \beta \log \frac{\pi_\theta(y_l|x)}{\pi_{ref}(y_l|x)} - \delta(x, y_w, y_l)\right)\right], \tag{17}$$

where the margin term $\delta(x, y_w, y_l)$ is defined as:

$$\delta(x, y_w, y_l) = \beta\mathbb{D}_{SeqKL}[x, y_l; \pi_{ref}||\pi_\theta] - \beta\mathbb{D}_{SeqKL}[x, y_w; \pi_{ref}||\pi_\theta], \tag{18}$$

$$= \beta \sum_{t=1}^{|y_l|} \mathbb{E}_{z\sim\pi_{ref}}[\log \frac{\pi_{ref}(z|[x, y_l^{<t}])}{\pi_\theta(z|[x, y_l^{<t}])}] - \beta \sum_{t=1}^{|y_w|} \mathbb{E}_{z\sim\pi_{ref}}[\log \frac{\pi_{ref}(z|[x, y_w^{<t}])}{\pi_\theta(z|[x, y_w^{<t}])}] \tag{19}$$

and $\mathbb{D}_{SeqKL}[x, y; \pi_{ref}||\pi_\theta]$ denotes the sequential KL divergence between $\pi_{ref}$ and $\pi_\theta$ along the sequence $y$ given $x$.

**Lemma 4.3** (Equivalence of Margin Terms)**.** *Let $\delta(x, y_w, y_l)$ denote the difference in sequential KL divergences between the reference policy $\pi_{ref}$ and the policy $\pi_\theta$ along the sequences $y_w$ and $y_l$, respectively, defined as:*

$$\delta(x, y_w, y_l) = \beta\mathbb{D}_{SeqKL}[x, y_l; \pi_{ref}||\pi_\theta] - \beta\mathbb{D}_{SeqKL}[x, y_w; \pi_{ref}||\pi_\theta],$$

*If we approximate the sequential KL divergences using the log-probability ratios of the sequences, then $\delta(x, y_w, y_l)$ simplifies to:*

$$\delta(x, y_w, y_l) \approx \beta\left(\log \frac{\pi_\theta(y_w|x)}{\pi_{ref}(y_w|x)} - \log \frac{\pi_\theta(y_l|x)}{\pi_{ref}(y_l|x)}\right).$$

*Consequently, $\delta(x, y_w, y_l)$ is approximately equivalent to the margin term $M(x, y_w, y_l)$ in the $\alpha$-DPO objective.*

Table 1: **AlpacaEval 2, Arena-Hard results across four settings.** "WR" denotes the raw win rate, "LC" the length-controlled win rate, and "SC" the style-controlled win rate. The best results are highlighted in bold, while the second-best are underlined.

| Method | Llama3-Instruct (8B) | | | | | Mistral-Instruct (7B) | | | | |
|---|---|---|---|---|---|---|---|---|---|---|
| | AlpacaEval 2 | | Arena-Hard | | | AlpacaEval 2 | | Arena-Hard | | |
| | LC (%) | WR (%) | SC (%) | LC (%) | WR (%) | LC (%) | WR (%) | SC (%) | LC (%) | WR (%) |
| SFT | 24.0 | 23.6 | 22.1 | 22.2 | 22.4 | 19.0 | 15.4 | 18.3 | 13.2 | 12.9 |
| DPO | 40.2 | **38.1** | 31.9 | 32.0 | 31.2 | 20.3 | 17.9 | 18.9 | 13.7 | 13.4 |
| IPO | 35.9 | 34.4 | 29.2 | 29.9 | 30.2 | 22.3 | 18.6 | 22.4 | 16.6 | 16.2 |
| CPO | 29.6 | 34.4 | 26.3 | 28.1 | 29.4 | 26.2 | 31.7 | 26.6 | 21.4 | **23.8** |
| KTO | 38.3 | 34.1 | 30.3 | 30.6 | 30.3 | 19.4 | 20.3 | 21.5 | 16.0 | 16.8 |
| ORPO | 31.6 | 29.8 | 26.6 | 26.6 | 26.3 | 24.0 | 23.0 | 24.4 | 18.5 | 18.6 |
| R-DPO | 40.3 | 37.3 | 33.1 | 32.9 | 32.9 | 21.4 | 22.2 | 18.7 | 14.0 | 13.8 |
| SimPO | 43.8 | 38.0 | 33.5 | 33.5 | 32.6 | 30.2 | 32.1 | 25.6 | 19.8 | 20.1 |
| $\alpha$-DPO | **46.6** | **38.1** | **34.1** | **34.2** | **33.3** | **32.3** | **32.6** | **27.2** | **21.5** | 21.5 |

| Method | Llama3-Instruct v0.2 (8B) | | | | | Gemma2-Instruct (9B) | | | | |
|---|---|---|---|---|---|---|---|---|---|---|
| | AlpacaEval 2 | | Arena-Hard | | | AlpacaEval 2 | | Arena-Hard | | |
| | LC (%) | WR (%) | SC (%) | LC (%) | WR (%) | LC (%) | WR (%) | SC (%) | LC (%) | WR (%) |
| SFT | 24.0 | 23.6 | 22.1 | 22.2 | 22.4 | 48.7 | 36.5 | 32.0 | 42.2 | 42.1 |
| DPO | 51.9 | 50.8 | 26.1 | 31.5 | 33.9 | 70.4 | **66.9** | 43.9 | 55.6 | 58.8 |
| IPO | 40.6 | 39.6 | **31.1** | 34.2 | 34.9 | 62.6 | 58.4 | 41.1 | 51.9 | 53.5 |
| CPO | 36.5 | 40.8 | 29.4 | 32.8 | 34.2 | 56.4 | 53.4 | 42.4 | 53.3 | 55.2 |
| KTO | 41.4 | 36.4 | 27.1 | 29.5 | 28.9 | 61.7 | 55.5 | 41.7 | 52.3 | 53.8 |
| ORPO | 36.5 | 33.1 | 28.8 | 30.8 | 30.4 | 56.2 | 46.7 | 35.1 | 45.3 | 46.2 |
| R-DPO | 51.6 | 50.7 | 29.2 | 34.3 | 35.0 | 68.3 | **66.9** | 45.1 | 55.9 | 57.9 |
| SimPO | 55.6 | 49.6 | 28.5 | 34.0 | 33.6 | 72.4 | 65.0 | 45.0 | 56.1 | 57.8 |
| $\alpha$-DPO | **58.7** | **51.1** | 30.8 | **36.3** | **35.7** | **73.4** | 66.1 | **48.6** | **59.3** | **60.8** |

The approximation leverages the assumption that the expectation over $z$ can be approximated by the log-probability ratios at the sequence level. Specifically, we recognize that the sequential KL divergence between $\pi_{\text{ref}}$ and $\pi_\theta$ along a sequence $y$ can be approximated by:

$$\mathbb{D}_{\text{SeqKL}}[x, y; \pi_{\text{ref}} || \pi_\theta] = \sum_{t=1}^{|y|} \mathbb{E}_{z \sim \pi_{\text{ref}}} \left[ \log \frac{\pi_{\text{ref}}(z|x, y^{<t})}{\pi_\theta(z|x, y^{<t})} \right] \approx \log \frac{\pi_{\text{ref}}(y|x)}{\pi_\theta(y|x)}.$$

This approximation reduces computational complexity by operating at the sequence level rather than the token level, making it particularly advantageous when dealing with long sequences or when the reference policy $\pi_{\text{ref}}$ is not well-calibrated at the token level. Applying this approximation to both $y_w$ and $y_l$, the difference $\delta(x, y_w, y_l)$ simplifies to the difference of log-probability ratios, thereby establishing the equivalence with the margin term in $\alpha$-DPO.

Lemma 4.3 highlights that the margin term $\delta(x, y_w, y_l)$, which represents the sequential KL divergence difference between preferred and rejected responses, can be directly mapped to the term $M(x, y_w, y_l)$ in $\alpha$-DPO. This mapping underscores the theoretical connection between the two approaches in terms of alignment control. While TDPO operates at the token level and provides fine-grained control, $\alpha$-DPO offers greater computational efficiency by operating at the sequence level without sacrificing performance. Moreover, the sequence-level approximation enhances robustness to token-level noise in $\pi_{\text{ref}}$, making $\alpha$-DPO particularly suited for scenarios where the reference policy may not be perfectly aligned. Refer to Appendix C.4, where we compare TDPO and $\alpha$-DPO.

# 5 EXPERIMENTS

In this section, we present the main results of our experiments, highlighting the superior performance of $\alpha$-DPO over existing methods on various benchmarks and ablation studies to analyze the impact of different components of $\alpha$-DPO.

## 5.1 EXPERIMENTS SETUP

**Models and training settings.** We optimize preferences using three model families: Llama3-8B (AI@Meta, 2024), Mistral2-7B (Jiang et al., 2023), and Gemma2-9B (Rivière et al., 2024), all in the Instruct setup. Following Meng et al. (2024), we utilize pre-trained instruction-tuned models

Table 2: **Ablation studies under Llama3-Instruct v0.2 and Mistral-Instruct settings.** We ablate each key design of $\alpha$-DPO and explore variants of the *implicit reference model* $\hat{\pi}_{\text{ref}}$.

| Method | Llama3-Instruct v0.2 (8B) | | | | | Mistral-Instruct (7B) | | | | |
|---|---|---|---|---|---|---|---|---|---|---|
| | AlpacaEval 2 | | Arena-Hard | | | AlpacaEval 2 | | Arena-Hard | | |
| | LC (%) | WR (%) | SC (%) | LC (%) | WR (%) | LC (%) | WR (%) | SC (%) | LC (%) | WR (%) |
| $U(\cdot\|x)$ | 55.6 | 49.6 | 28.5 | 34.0 | 33.6 | 30.2 | 32.1 | 25.6 | 19.8 | 20.1 |
| $U(\cdot\|x)\,(\pi_\theta(\cdot\|x)/\pi_{\text{ref}}(\cdot\|x))^\alpha$ | 58.7 | 51.1 | 30.8 | 36.3 | 35.7 | 32.3 | 32.6 | 27.2 | 21.5 | 21.5 |
| w/o Normalization | 56.5 | 49.7 | 23.1 | 28.4 | 27.7 | 32.1 | 33.1 | 25.2 | 19.7 | 19.6 |
| w/o sg | 2.7 | 3.7 | 7.7 | 5.4 | 6.3 | 27.2 | 27.7 | 25.8 | 20.3 | 20.7 |
| $\gamma = 0$ | 51.2 | 44.9 | 30.0 | 34.5 | 33.3 | 31.9 | 31.3 | 24.2 | 19.6 | 19.3 |
| $U(\cdot\|x)\,(\pi_\theta(\cdot\|x))^\alpha$ | 57.2 | 50.4 | 27.6 | 33.5 | 32.9 | 31.6 | 34.1 | 26.9 | 21.3 | 21.5 |
| $U(\cdot\|x)\,(\pi_{\text{ref}}(\cdot\|x))^\alpha$ | 56.3 | 49.5 | 29.0 | 34.3 | 33.5 | 28.6 | 30.9 | 25.5 | 20.1 | 20.3 |
| $U(\cdot\|x)\,(1/\pi_{\text{ref}}(\cdot\|x))^\alpha$ | 56.3 | 49.2 | 29.0 | 34.4 | 33.8 | 32.2 | 33.1 | 26.0 | 20.7 | 20.6 |

(meta-llama/Meta-Llama-3-8B-Instruct, mistralai/Mistral-7B-Instruct-v0.2, google/gemma-2-9b-it) as SFT models. For a fair comparison, we use the same training data as SimPO: princeton-nlp/llama3-ultrafeedback-armorm[1], princeton-nlp/mistral-instruct-ultrafeedback[2], and princeton-nlp/gemma2-ultrafeedback-armorm [3] for Llama3-8B, Mistral2-7B, and Gemma2-9B, respectively. Additionally, the v0.2 Llama3-Instruct setup uses RLHFlow/ArmoRM-Llama3-8B-v0.1 (Wang et al., 2024b) as the reward model for ranking generated data, significantly enhancing performance. These configurations represent state-of-the-art methods, positioning our models among the top performers on various leaderboards.

**Evaluation benchmarks.** We evaluate our models using two widely recognized open-ended instruction-following benchmarks: AlpacaEval 2 (Li et al., 2023) and Arena-Hard (Li et al., 2024). These benchmarks assess the models' conversational abilities across a diverse range of queries and are extensively used by the research community. For AlpacaEval 2, we report the length-controlled win rate (LC) and raw win rate (WR). For Arena-Hard, we provide the win rate (WR), length-controlled win rate (LC), and style-controlled win rate (SC) compared to baseline models. Note that style significantly impacts performance on these leaderboards.

**Baselines.** We compare $\alpha$-DPO with several state-of-the-art preference optimization methods: DPO (Rafailov et al., 2023), SimPO (Meng et al., 2024), IPO (Azar et al., 2023), CPO (Xu et al., 2024), KTO (Ethayarajh et al., 2024), ORPO (Hong et al., 2024), and R-DPO (Park et al., 2024). We also include the SFT model as a baseline. We thoroughly tune the hyperparameters for each baseline and report the best performance. Further details can be found in Appendix C.1.

## 5.2 MAIN RESULTS

$\alpha$**-DPO consistently outperforms existing preference optimization methods.** As shown in Table 1, while all preference optimization algorithms improve over the SFT baseline, $\alpha$-DPO achieves superior performance compared to existing methods specifically on the AlpacaEval 2 LC metric. These significant improvements highlight the robustness and effectiveness of $\alpha$-DPO. Specifically, $\alpha$-DPO outperforms the best baseline by an average of 3 percentage points in AlpacaEval 2 LC win rate. Furthermore, on benchmarks such as Arena-Hard, $\alpha$-DPO achieves state-of-the-art or second-best results, demonstrating its competitiveness across different evaluation settings.

**Impact of Metrics on Leaderboard Rankings.** While both benchmarks are widely used, the standard win rate (WR) metric shows poor separability among different methods, making it challenging to distinguish their relative performance. Minor differences in WR may stem from biases towards generating detailed or aesthetically pleasing responses, aligning with observations by Dubois et al. (2024) and Chen et al. (2024a). In contrast, the length-controlled (LC) and style-controlled (SC) win rates offer more reliable and interpretable metrics, as they reduce the influence of verbosity and stylistic biases, thereby better reflecting true performance.

**The importance of the design on the implicit reference model.** As the core contribution of this work is to propose a novel reference model $\hat{\pi}_{\text{ref}}(y|x) = U(y|x)\,(\pi_\theta(y|x)/\pi_{\text{ref}}(y|x))^\alpha$, we also evaluate other variants of the reference model. Specifically, we compare $\alpha$-DPO with three variants: (1) $\hat{\pi}_{\text{ref}}(y|x) = U(y|x)\,(\pi_\theta(y|x))^\alpha$, (2) $\hat{\pi}_{\text{ref}}(y|x) = U(y|x)\,(\pi_{\text{ref}}(y|x))^\alpha$, and (3)

---

[1]https://huggingface.co/datasets/princeton-nlp/llama3-ultrafeedback-armorm

[2]https://huggingface.co/datasets/princeton-nlp/mistral-instruct-ultrafeedback

[3]https://huggingface.co/datasets/princeton-nlp/gemma2-ultrafeedback-armorm

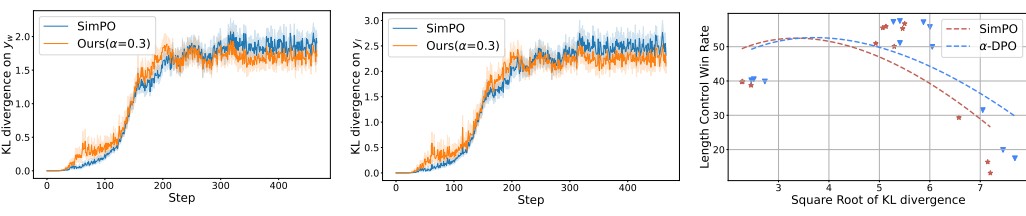

(a) KL divergence of $y_w$ over steps.  (b) KL divergence of $y_l$ over steps.  (c) LC and KL divergence.

Figure 3: Analysis of KL divergence and LC trade-off. (a) KL divergence for chosen samples ($y_w$), (b) KL divergence for rejected samples ($y_l$), and (c) relationship between LC and KL divergence.

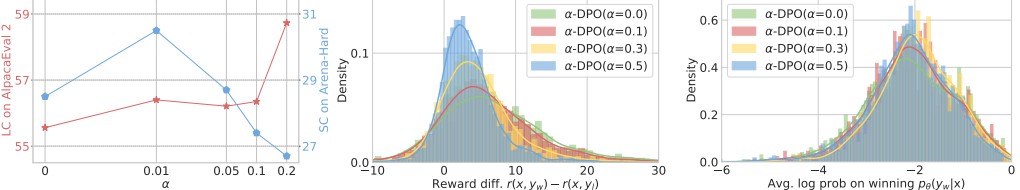

Figure 4: Impact of $\alpha$ on (a) LC and SC win rate, (b) reward difference distribution, and (c) log-likelihood distribution of chosen responses in $\alpha$-DPO.

$\hat{\pi}_{\text{ref}}(y|x) = U(y|x)\,(1/\pi_{\text{ref}}(y|x))^{\alpha}$. As shown in Table 2, most of the variants perform better than SimPO ($\hat{\pi}_{\text{ref}}(y|x) = U(y|x)$), which demonstrates the importance of adaptive margin between pairs. Besides, our proposed reference model consistently outperforms other variants, indicating the effectiveness of the proposed design.

**All key designs in $\alpha$-DPO are crucial.** To further analyze the impact of different components in $\alpha$-DPO, we conduct ablation studies by removing key components from $\alpha$-DPO. As shown in Table 2, removing normalization, stop gradient, or setting $\gamma = 0$ all lead to significant performance drops, highlighting the importance of these components in $\alpha$-DPO.

## 5.3 KL DIVERGENCE CONTROL IN $\alpha$-DPO

**Outstanding Performance and Lower KL.** As noted in Rafailov et al. (2023); Zeng et al. (2024), it is crucial to consider both performance and KL divergence when comparing algorithms. A slightly higher win rate accompanied by a significantly higher KL divergence is often not desirable. In line with the design principles of TDPO, we implemented SimPO and $\alpha$-DPO. Figure 3a 3b presents the KL divergence curves. The results indicate that as $\alpha$ increases, the KL divergence of $\alpha$-DPO remains stable or even decreases slightly when compared to SimPO. This demonstrates that $\alpha$-DPO not only achieves superior performance but also maintains a lower KL divergence, indicating a better balance between alignment and control of KL divergence during the training process.

**Mitigating Over-Optimization.** Over-optimization, as described by Gao et al. (2023) and Rafailov et al. (2024), refers to a phenomenon where model performance exhibits a hump-shaped pattern across different targets: beyond an optimal point, further increasing the KL budget results in diminishing returns. To investigate this, we evaluate SimPO and $\alpha$-DPO at four intermediate checkpoints, corresponding to different KL budgets. As illustrated in Figure 3c, it is intriguing that while the performance of our approach does decrease with increasing KL budget, the decline is relatively modest. This indicates that our method effectively mitigates the issue of over-optimization.

## 5.4 THE IMPACT OF $\alpha$ IN $\alpha$-DPO

**Effect of $\alpha$ on Performance.** We investigated how the parameter $\alpha$ in $\alpha$-DPO impacts the win rate on AlpacaEval 2 and Arena-Hard. The results, as shown in Figure 4 (a), indicate that the style-control win rate on Arena-Hard initially increases and then decreases with increasing $\alpha$. In contrast, the length-control win rate on AlpacaEval 2 exhibits a consistently increasing trend. This suggests that the optimal value of $\alpha$ varies depending on the evaluation benchmarks. Further experiments refer to Appendix C.3.

**Impact of $\alpha$ on the reward distribution.** We visualize the distribution of the learned reward margin $r(x, y_w) - r(x, y_l)$ and the log likelihood of the chosen response $\log \pi_\theta(y_w|x)$ under different $\alpha$

values in Figure 4 (b,c). Decreasing $\alpha$ results in a flatter reward margin, while the log likelihood distribution remains relatively unchanged. Conversely, in SimPO (*cf.* Figure 6), increasing $\gamma$ yields a flatter reward margin distribution but at the cost of also flattening the log likelihood distribution, which undesirably lowers the log likelihood of positive samples. This indicates that $\alpha$-DPO can better balance the relationship between the reward margin and log likelihood.

## 6 RELATED WORK

**Reinforcement learning from human feedback.** RLHF is a technique that aligns large language models with human preferences and values (Christiano et al., 2017; Ziegler et al., 2019; Ouyang et al., 2022; Azar et al., 2023). Traditional RLHF can be divided into three stages: supervised fine-tuning (Zhou et al., 2023; Taori et al., 2023; Geng et al., 2023; Conover et al., 2023; Köpf et al., 2023; Ding et al., 2023), reward modeling (Gao et al., 2023; Luo et al., 2023; Chen et al., 2024b; Lightman et al., 2023; Havrilla et al., 2024; Lambert et al., 2024), and policy optimization (Schulman et al., 2017; Anthony et al., 2017). In the third stage, Proximal Policy Optimization (PPO) is a widely used algorithm. Additionally, Xiong et al. (2023) proposed efficient algorithms for the reverse-KL regularized contextual bandit framework in RLHF. Ye et al. (2024) introduced provably efficient algorithms for KL-regularized Nash-Learning from Human Feedback (NLHF). Furthermore, Ji et al. (2024b) developed an active-query-based PPO algorithm with specific regret bounds and query complexity.

**Offline direct preference optimization.** Several alternative preference optimization objectives have been proposed in addition to DPO (Rafailov et al., 2023). IPO (Azar et al., 2023) addresses the overfitting issues associated with DPO. ORPO (Hong et al., 2024) and SimPO (Meng et al., 2024) aim to eliminate the dependence on a reference model. R-DPO (Park et al., 2024) focuses on mitigating exploitation based on sequence length. KTO (Ethayarajh et al., 2024) deals with preference optimization when data are not pairwise. CPO (Xu et al., 2024) and $\beta$-DPO(Wu et al., 2024) emphasize the quality of preference data. Another line of research explores comparisons among more than two instances (Dong et al., 2023; Liu et al., 2024a; Song et al., 2024; Yuan et al., 2023).

**Online direct preference optimization.** Offline direct preference optimization methods are simple but rely on preference data collected offline. RLHF methods interact online with the language model being aligned but require policy gradients. In contrast, online direct preference optimization methods combine the advantages of both approaches. Yuan et al. (2024) proposed a "self-rewarding" approach in which the policy being aligned provides online feedback to itself. Alternatively, OAIF (Guo et al., 2024) is a novel online preference optimization method that can leverage feedback from any LLM, including those stronger than the LLM being aligned. Swamy et al. (2024) also concurrently investigate the importance of online preference but still rely on reward models (RMs). SELMA (Zhang et al., 2024) improves exploration efficiency by selectively favoring responses with high potential rewards rather than indiscriminately sampling unseen responses.

## 7 DISCUSSION

**Conclusion.** We proposed $\alpha$-DPO, an adaptive preference optimization method that improves LLM alignment by introducing a dynamic reward margin based on instance-specific differences. $\alpha$-DPO addresses limitations in previous methods like DPO and SimPO by balancing alignment and diversity through KL divergence control. Our theoretical guarantees and empirical results show that $\alpha$-DPO consistently outperforms baselines on benchmarks like AlpacaEval 2 and Arena-Hard, with significant improvements in win rates, establishing it as a robust solution for LLM fine-tuning.

**Limitations and Future Work.** While $\alpha$-DPO enhances performance, it introduces an additional hyperparameter, $\alpha$, requiring manual tuning. Future work could focus on developing an adaptive approach to automatically adjust this parameter. Additionally, although we show $\alpha$-DPO's theoretical equivalence to online methods, it remains an offline approach. Extending it to online learning would allow real-time adaptation, broadening its application in interactive environments. Lastly, we observed that different benchmarks, such as AlpacaEval 2 and Arena-Hard, require distinct parameter settings for optimal performance. Investigating a more generalized approach that adapts effectively across multiple benchmarks would further improve the model's versatility.

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

# A APPENDIX

## A.1 PROOF OF THEOREM 3.1

**Theorem 3.1.** *Let $U(y|x)$ denote a uniform distribution over the vocabulary for a given input $x$, replacing $\pi_{ref}(y|x)$ in the DPO loss function. Then, the DPO loss function simplifies to:*

$$\mathcal{L}(\pi_\theta; U) = -\mathbb{E}_{(x,y_w,y_l)\sim\mathcal{D}}\left[\log\sigma\left(\beta\left(\log\pi_\theta(y_w|x) - \log\pi_\theta(y_l|x)\right) - \gamma\right)\right], \quad (6)$$

*where $\gamma = \beta\left(\log U(y_w|x) - \log U(y_l|x)\right)$ is a constant. Under a length-normalized reward formulation, this loss function becomes:*

$$\mathcal{L}_{LN}(\pi_\theta; U) = -\mathbb{E}_{(x,y_w,y_l)\sim\mathcal{D}}\left[\log\sigma\left(\frac{\beta}{|y_w|}\log\pi_\theta(y_w|x) - \frac{\beta}{|y_l|}\log\pi_\theta(y_l|x) - \gamma\right)\right]. \quad (7)$$

*Therefore, SimPO can be interpreted as a special case of DPO where the reference model is a uniform distribution.*

*Proof.* Let $U(y|x)$ denote a uniform distribution over the vocabulary $\mathcal{V}$ for a given input $x$. Specifically, for any sequence $y$, the uniform distribution is defined as:

$$U(y|x) = \prod_{t=1}^{|y|}\frac{1}{|\mathcal{V}|} = \left(\frac{1}{|\mathcal{V}|}\right)^{|y|}.$$

Consider the DPO loss function:

$$\mathcal{L}_{\text{DPO}}(\pi_\theta; \pi_{\text{ref}}) = -\mathbb{E}_{(x,y_w,y_l)\sim\mathcal{D}}\left[\log\sigma\left(\beta\log\frac{\pi_\theta(y_w|x)}{\pi_\theta(y_l|x)} - \beta\log\frac{\pi_{\text{ref}}(y_w|x)}{\pi_{\text{ref}}(y_l|x)}\right)\right].$$

By substituting $\pi_{\text{ref}} = U$, the term involving the reference policy simplifies to:

$$\beta\log\frac{\pi_{\text{ref}}(y_w|x)}{\pi_{\text{ref}}(y_l|x)} = \beta\left(\log U(y_w|x) - \log U(y_l|x)\right) = \gamma,$$

where $\gamma$ is a constant. This constancy arises because $y_w$ and $y_l$ are chosen from distinct subsets of the vocabulary, ensuring that $\log U(y_w|x) - \log U(y_l|x)$ does not depend on the lengths of the sequences but is instead determined by the fixed probabilities of the respective subsets. Consequently, $\gamma$ remains fixed across all samples in $\mathcal{D}$.

Substituting back into the DPO loss function, we obtain:

$$\mathcal{L}(\pi_\theta; U) = -\mathbb{E}_{(x,y_w,y_l)\sim\mathcal{D}}\left[\log\sigma\left(\beta\left(\log\pi_\theta(y_w|x) - \log\pi_\theta(y_l|x)\right) - \gamma\right)\right].$$

Under the length-normalized reward formulation, the rewards are adjusted by the lengths of the sequences $y_w$ and $y_l$. This normalization yields:

$$\mathcal{L}_{\text{LN}}(\pi_\theta; U) = -\mathbb{E}_{(x,y_w,y_l)\sim\mathcal{D}}\left[\log\sigma\left(\frac{\beta}{|y_w|}\log\pi_\theta(y_w|x) - \frac{\beta}{|y_l|}\log\pi_\theta(y_l|x) - \gamma\right)\right].$$

Here, $\gamma$ remains a fixed constant since it is derived from the uniform distribution over distinct vocabulary subsets corresponding to $y_w$ and $y_l$.

Comparing this with the SimPO loss function:

$$\mathcal{L}_{\text{SimPO}}(\pi_\theta) = -\mathbb{E}_{(x,y_w,y_l)\sim\mathcal{D}}\left[\log\sigma\left(\frac{\beta}{|y_w|}\log\pi_\theta(y_w|x) - \frac{\beta}{|y_l|}\log\pi_\theta(y_l|x) - \gamma\right)\right],$$

it is evident that:

$$\mathcal{L}_{\text{LN}}(\pi_\theta; U) = \mathcal{L}_{\text{SimPO}}(\pi_\theta).$$

Thus, when the reference policy $\pi_{\text{ref}}$ is a uniform distribution over distinct vocabulary subsets for $y_w$ and $y_l$, the DPO loss function simplifies to the SimPO loss function with $\gamma$ being a fixed constant. This establishes that SimPO is a special case of DPO under the specified conditions. □

## A.2 PROOF OF LEMMA 4.2

**Lemma 4.2** (Tight bound between $\alpha$-DPO and online SimPO loss). *For any policy model $\pi_\theta$ and reference model $\pi_{ref}$, there exists a sufficiently small $\alpha > 0$ such that the following inequalities hold:*

$$|\mathcal{L}_{SimPO}^{online}(\pi_\theta, \pi_{ref}) - \mathcal{L}_{\alpha\text{-}DPO}(\pi_\theta, \pi_{ref})| \leq \varepsilon(\alpha),$$

*where*

$$\varepsilon(\alpha) = \mathbb{E}_{\pi_{ref}}\left[\alpha|B|\left|\log \sigma(A) - \sigma(A) + 1\right|\right],$$

$A = \frac{\beta}{|y_w|}\log \pi_\theta(y_w|x) - \frac{\beta}{|y_l|}\log \pi_\theta(y_l|x) - \gamma$, *and* $B = \log \frac{\pi_\theta(y_w|x)}{\pi_{ref}(y_w|x)} - \log \frac{\pi_\theta(y_l|x)}{\pi_{ref}(y_l|x)}$.

*Proof.* To establish the tight bound between $L_2 = \mathcal{L}_{\alpha\text{-}DPO}(\pi_\theta, \pi_{ref})$ and $L_1 = \mathcal{L}_{SimPO}^{online}(\pi_\theta, \pi_{ref})$, we proceed as follows.

The online SimPO loss $L_1$ is defined as:

$$L_1 = -\mathbb{E}_{\pi_{ref}}\left[\log \sigma\left(\frac{\beta}{|y_w|}\log \pi_\theta(y_w|x) - \frac{\beta}{|y_l|}\log \pi_\theta(y_l|x) - \gamma\right) \cdot w(y_w, y_l|x)\right],$$

where the importance weight $w(y_w, y_l|x)$ is:

$$w(y_w, y_l|x) = \frac{\pi_{\theta,old}(y_w|x)}{\pi_{ref}(y_w|x)} \cdot \frac{\pi_{ref}(y_l|x)}{\pi_{\theta,old}(y_l|x)}.$$

Under the relationship between the old policy $\pi_{\theta,old}$ and the current policy $\pi_\theta$:

$$\pi_{\theta,old}(y|x) = C(y|x)\left(\frac{\pi_\theta(y|x)}{\pi_{ref}(y|x)}\right)^\alpha \pi_{ref}(y|x),$$

the importance weight $w(y_w, y_l|x)$ simplifies to:

$$w(y_w, y_l|x) = \frac{C(y_w|x)}{C(y_l|x)}\left(\frac{\pi_\theta(y_w|x)}{\pi_{ref}(y_w|x)} \cdot \frac{\pi_{ref}(y_l|x)}{\pi_\theta(y_l|x)}\right)^\alpha.$$

Assuming $\alpha$ is sufficiently small and $C(y|x)$ varies smoothly, we approximate:

$$\frac{C(y_w|x)}{C(y_l|x)} \approx 1,$$

leading to:

$$w(y_w, y_l|x) \approx \left(\frac{\pi_\theta(y_w|x)}{\pi_{ref}(y_w|x)} \cdot \frac{\pi_{ref}(y_l|x)}{\pi_\theta(y_l|x)}\right)^\alpha = e^{\alpha B},$$

where $B = \log \frac{\pi_\theta(y_w|x)}{\pi_{ref}(y_w|x)} - \log \frac{\pi_\theta(y_l|x)}{\pi_{ref}(y_l|x)}$.

Define $A = \frac{\beta}{|y_w|}\log \pi_\theta(y_w|x) - \frac{\beta}{|y_l|}\log \pi_\theta(y_l|x) - \gamma$. Substituting, $L_1$ becomes:

$$L_1 = -\mathbb{E}_{\pi_{ref}}\left[\log \sigma(A) \cdot e^{\alpha B}\right].$$

Using a Taylor series expansion, we approximate:

$$w(y_w, y_l|x) = e^{\alpha B} \approx 1 + \alpha B + \frac{\alpha^2 B^2}{2} + \mathcal{O}(\alpha^3).$$

Similarly, we expand $\log \sigma(A - \alpha B)$ around $A$:

$$\log \sigma(A - \alpha B) \approx \log \sigma(A) - \alpha B(1 - \sigma(A)) + \frac{\alpha^2 B^2}{2}(\sigma(A) - \sigma(A)^2) + \mathcal{O}(\alpha^3),$$

where:

$$\frac{d}{dA}\log \sigma(A) = 1 - \sigma(A), \quad \frac{d^2}{dA^2}\log \sigma(A) = \sigma(A) - \sigma(A)^2.$$

For $L_1$, substituting the expansion of $w(y_w, y_l|x)$:

$$L_1 = -\mathbb{E}_{\pi_{\text{ref}}}\left[\log \sigma(A) \cdot w(y_w, y_l|x)\right]$$

$$\approx -\mathbb{E}_{\pi_{\text{ref}}}\left[\log \sigma(A)(1 + \alpha B + \frac{\alpha^2 B^2}{2})\right]$$

$$= -\mathbb{E}_{\pi_{\text{ref}}}[\log \sigma(A)] - \alpha\mathbb{E}_{\pi_{\text{ref}}}[B \log \sigma(A)] - \frac{\alpha^2}{2}\mathbb{E}_{\pi_{\text{ref}}}[B^2 \log \sigma(A)] + \mathcal{O}(\alpha^3).$$

For $L_2$, substituting the expansion of $\log \sigma(A - \alpha B)$:

$$L_2 \approx -\mathbb{E}_{\pi_{\text{ref}}}\left[\log \sigma(A) - \alpha B(1 - \sigma(A)) + \frac{\alpha^2 B^2}{2}(\sigma(A) - \sigma(A)^2)\right]$$

$$= -\mathbb{E}_{\pi_{\text{ref}}}[\log \sigma(A)] + \alpha\mathbb{E}_{\pi_{\text{ref}}}[B(1 - \sigma(A))] - \frac{\alpha^2}{2}\mathbb{E}_{\pi_{\text{ref}}}[B^2(\sigma(A) - \sigma(A)^2)] + \mathcal{O}(\alpha^3).$$

The difference $L_1 - L_2$ is:

$$L_1 - L_2 = -\alpha\mathbb{E}_{\pi_{\text{ref}}}\left[B(\log \sigma(A) + 1 - \sigma(A))\right]$$

$$- \frac{\alpha^2}{2}\mathbb{E}_{\pi_{\text{ref}}}\left[B^2(\log \sigma(A) + \sigma(A) - \sigma(A)^2)\right] + \mathcal{O}(\alpha^3).$$

The magnitude of the difference is bounded by:

$$|L_1 - L_2| \leq \alpha\mathbb{E}_{\pi_{\text{ref}}}\left[|B| \cdot |\log \sigma(A) - \sigma(A) + 1|\right] + \frac{\alpha^2}{2}\mathbb{E}_{\pi_{\text{ref}}}\left[B^2 \cdot |\log \sigma(A) + \sigma(A) - \sigma(A)^2|\right] + \mathcal{O}(\alpha^3).$$

By defining:

$$\varepsilon(\alpha) = \mathbb{E}_{\pi_{\text{ref}}}\left[\alpha|B| \cdot |\log \sigma(A) - \sigma(A) + 1|\right],$$

and neglecting higher-order terms for small $\alpha$, we obtain:

$$|L_1 - L_2| \leq \varepsilon(\alpha).$$

$\square$

### A.3 Proof of Lemma 4.3

**Lemma 4.3** (Equivalence of Margin Terms). *Let $\delta(x, y_w, y_l)$ denote the difference in sequential KL divergences between the reference policy $\pi_{ref}$ and the policy $\pi_\theta$ along the sequences $y_w$ and $y_l$, respectively, defined as:*

$$\delta(x, y_w, y_l) = \beta\mathbb{D}_{SeqKL}[x, y_l; \pi_{ref}||\pi_\theta] - \beta\mathbb{D}_{SeqKL}[x, y_w; \pi_{ref}||\pi_\theta],$$

*If we approximate the sequential KL divergences using the log-probability ratios of the sequences, then $\delta(x, y_w, y_l)$ simplifies to:*

$$\delta(x, y_w, y_l) \approx \beta\left(\log \frac{\pi_\theta(y_w|x)}{\pi_{ref}(y_w|x)} - \log \frac{\pi_\theta(y_l|x)}{\pi_{ref}(y_l|x)}\right).$$

*Consequently, $\delta(x, y_w, y_l)$ is approximately equivalent to the margin term $M(x, y_w, y_l)$ in the $\alpha$-DPO objective.*

*Proof of Lemma 4.3.* We begin by expanding the definition of $\delta(x, y_w, y_l)$:

$$\delta(x, y_w, y_l) = \beta\mathbb{D}_{\text{SeqKL}}[x, y_l; \pi_{\text{ref}}||\pi_\theta] - \beta\mathbb{D}_{\text{SeqKL}}[x, y_w; \pi_{\text{ref}}||\pi_\theta]$$

Expanding each sequential KL divergence, we have:

$$\mathbb{D}_{\text{SeqKL}}[x, y; \pi_{\text{ref}}||\pi_\theta] = \sum_{t=1}^{|y|} \mathbb{E}_{z \sim \pi_{\text{ref}}}\left[\log \frac{\pi_{\text{ref}}(z \mid [x, y^{<t}])}{\pi_\theta(z \mid [x, y^{<t}])}\right]$$

Substituting this into the expression for $\delta$, we obtain:

$$\delta(x, y_w, y_l) = \beta \sum_{t=1}^{|y_l|} \mathbb{E}_{z \sim \pi_{\text{ref}}} \left[ \log \frac{\pi_{\text{ref}}(z \mid [x, y_l^{<t}])}{\pi_\theta(z \mid [x, y_l^{<t}])} \right] - \beta \sum_{t=1}^{|y_w|} \mathbb{E}_{z \sim \pi_{\text{ref}}} \left[ \log \frac{\pi_{\text{ref}}(z \mid [x, y_w^{<t}])}{\pi_\theta(z \mid [x, y_w^{<t}])} \right]$$

Under the assumption that the reference policy $\pi_{\text{ref}}$ has large errors, we approximate the expectation $\mathbb{E}_{z \sim \pi_{\text{ref}}}$ with a uniform distribution. This approximation simplifies each expectation term as follows:

$$\sum_{t=1}^{|y|} \mathbb{E}_{z \sim \pi_{\text{ref}}} \left[ \log \frac{\pi_{\text{ref}}(z \mid [x, y^{<t}])}{\pi_\theta(z \mid [x, y^{<t}])} \right] \approx \log \frac{\pi_\theta(y \mid x)}{\pi_{\text{ref}}(y \mid x)}$$

Applying this approximation to both sequential KL divergence terms, we obtain:

$$\delta(x, y_w, y_l) \approx \beta \left( \log \frac{\pi_\theta(y_l \mid x)}{\pi_{\text{ref}}(y_l \mid x)} - \log \frac{\pi_\theta(y_w \mid x)}{\pi_{\text{ref}}(y_w \mid x)} \right)$$

This expression can be rewritten as:

$$\delta(x, y_w, y_l) \approx \beta \left( \log \frac{\pi_\theta(y_w \mid x)}{\pi_{\text{ref}}(y_w \mid x)} - \log \frac{\pi_\theta(y_l \mid x)}{\pi_{\text{ref}}(y_l \mid x)} \right) = M(x, y_w, y_l)$$

where $M(x, y_w, y_l)$ is the margin term defined in the $\alpha$-DPO objective. Thus, we have shown that:

$$\delta(x, y_w, y_l) \approx M(x, y_w, y_l)$$

This completes the proof. $\qquad\square$

## B    THE MOTIVATION FOR THE PROPOSED $\hat{\pi}_{\text{REF}}(y|x)$

The motivation for the proposed reference policy $\hat{\pi}_{\text{ref}}(y|x)$ can be clarified as follows:

- **Utility Theory Perspective:** The proposed $\hat{\pi}_{\text{ref}}(y|x)$ is designed with the uniform distribution $U(y|x)$ as a baseline. The term $\left( \frac{\pi_\theta(y|x)}{\pi_{\text{ref}}(y|x)} \right)^\alpha$ dynamically adjusts the reward margin by balancing contributions from the policy and reference models. This mechanism can be interpreted through the lens of utility theory as relative attractiveness, enabling adaptive instance-specific reward modeling.

- **Gradient Perspective** By introducing $\hat{\pi}_{\text{ref}}(y|x)$, the framework mitigates the label flipping issues found in DPO or SimPO. In the SimPO framework, the gradient is expressed as:

$$\nabla_\theta \mathcal{L}_{\text{SimPO}}(\pi_\theta) = -\beta \mathbb{E}_{(x, y_w, y_l) \sim \mathcal{D}} \left[ s_\theta \left( \frac{1}{|y_w|} \nabla_\theta \log \pi_\theta(y_w|x) - \frac{1}{|y_l|} \nabla_\theta \log \pi_\theta(y_l|x) \right) \right],$$

where $s_\theta = \sigma \left( \frac{\beta}{|y_l|} \log \pi_\theta(y_l|x) - \frac{\beta}{|y_w|} \log \pi_\theta(y_w|x) + \gamma \right)$.

This formulation may amplify weights when the reward estimate is incorrect. By contrast, under $\alpha$-DPO:

$$s_\theta = \sigma \left( \frac{\beta}{|y_l|} \log \pi_\theta(y_l|x) - \frac{\beta}{|y_w|} \log \pi_\theta(y_w|x) + \gamma + \alpha M(x, y_w, y_l) \right),$$

the additional $\alpha M(x, y_w, y_l)$ component increases weight when the reward estimate is accurate, ensuring a more robust reward signal.

- **Motivational Core** The central goal of the proposed $\alpha$-DPO is to address the unreliability of the reference policy, as outlined in Section 3.1. By integrating the policy model into the reference model design, the quality of the reference model is enhanced, improving fine-tuning performance. Similar concepts have been explored in recent works (Gorbatovski et al., 2024; Liu et al., 2024b).

# C EXPERIMENTS

## C.1 IMPLEMENTATION DETAILS

We observed that the performance of various methods is highly sensitive to model parameters and learning rates. To ensure a fair comparison, we conducted a hyperparameter search as specified in the respective papers. The specific search ranges are detailed in Table 3. Furthermore, due to recent updates to both Llama3-8b and Instruct-7b models, we had to re-implement SimPO as the original results were no longer directly applicable.

**Training hyperparameters.** For other parameters, we used a consistent batch size of 128 across all methods. The learning rate was searched within the range of [3e-7, 5e-7, 8e-7, 1e-6], and all models were trained for a single epoch with a cosine learning rate schedule and a 10% warmup phase. Adam was used as the optimizer (Kingma & Ba, 2014). Additionally, the maximum sequence length was set to 2048.

Table 3: Various preference optimization objectives and hyperparameter search range.

| Method | Objective | Hyperparameter |
|---|---|---|
| DPO (Rafailov et al., 2023) | $-\log \sigma \left( \beta \log \frac{\pi_\theta(y_w|x)}{\pi_{\text{ref}}(y_w|x)} - \beta \log \frac{\pi_\theta(y_l|x)}{\pi_{\text{ref}}(y_l|x)} \right)$ | $\beta \in [0.01, 0.05, 0.1]$ |
| IPO (Azar et al., 2023) | $\left( \log \frac{\pi_\theta(y_w|x)}{\pi_{\text{ref}}(y_w|x)} - \log \frac{\pi_\theta(y_l|x)}{\pi_{\text{ref}}(y_l|x)} - \frac{1}{2\tau} \right)^2$ | $\tau \in [0.01, 0.1, 0.5, 1.0]$ |
| CPO (Xu et al., 2024) | $-\log \sigma \left( \beta \log \pi_\theta(y_w|x) - \beta \log \pi_\theta(y_l|x) \right) - \lambda \log \pi_\theta(y_w|x)$ | $\alpha = 1.0, \ \beta \in [0.01, 0.05, 0.1]$ |
| KTO (Ethayarajh et al., 2024) | $-\lambda_w \sigma \left( \beta \log \frac{\pi_\theta(y_w|x)}{\pi_{\text{ref}}(y_w|x)} - z_{\text{ref}} \right) + \lambda_l \sigma \left( z_{\text{ref}} - \beta \log \frac{\pi_\theta(y_l|x)}{\pi_{\text{ref}}(y_l|x)} \right),$ where $z_{\text{ref}} = \mathbb{E}_{(x,y)\sim\mathcal{D}} \left[ \beta \text{KL} \left( \pi_\theta(y|x) || \pi_{\text{ref}}(y|x) \right) \right]$ | $\lambda_l = \lambda_w = 1.0$ $\beta \in [0.01, 0.05, 0.1]$ |
| ORPO (Hong et al., 2024) | $-\log p_\theta(y_w|x) - \lambda \log \sigma \left( \log \frac{p_\theta(y_w|x)}{1-p_\theta(y_w|x)} - \log \frac{p_\theta(y_l|x)}{1-p_\theta(y_l|x)} \right),$ where $p_\theta(y|x) = \exp\left( \frac{1}{|y|} \log \pi_\theta(y|x) \right)$ | $\lambda \in [0.1, 0.5, 1.0, 2.0]$ |
| R-DPO (Park et al., 2024) | $-\log \sigma \left( \beta \log \frac{\pi_\theta(y_w|x)}{\pi_{\text{ref}}(y_w|x)} - \beta \log \frac{\pi_\theta(y_l|x)}{\pi_{\text{ref}}(y_l|x)} - (\alpha|y_w| - \alpha|y_l|) \right)$ | $\alpha \in [0.05, 0.1, 0.5, 1.0]$ $\beta \in [0.01, 0.05, 0.1]$ |
| SimPO (Meng et al., 2024) | $-\log \sigma \left( \frac{\beta}{|y_w|} \log \pi_\theta(y_w|x) - \frac{\beta}{|y_l|} \log \pi_\theta(y_l|x) - \gamma \right)$ | $\beta \in [2.0, 4.0, 6.0, 8.0]$ $\gamma \in [0.3, 0.5, 1.0, 1.2, 1.4, 1.6]$ |
| $\alpha$-DPO | $-\log \sigma \left( u(x, y_w, y_l) - \text{sg}\left[ \gamma + \alpha M^*(x, y_w, y_l) \right] \right)$ where $u(x, y_w, y_l) = \frac{\beta}{|y_w|} \log \pi_\theta(y_w|x) - \frac{\beta}{|y_l|} \log \pi_\theta(y_l|x)$ | $\beta \in [2.5, 10.0], \ \gamma \in [0.1, 0.3, 0.5]$ $\alpha \in [1e-2, 5e-2, 0.1, 0.2]$ |

Table 4: The hyperparameter values in $\alpha$-DPO used for each training setting.

| Setting | $\beta$ | $\gamma$ | $\alpha$ | Learning rate |
|---|---|---|---|---|
| **Mistral-Instruct** | 2.5 | 0.15 | 5e-2 | 6e-7 |
| **Llama3-Instruct** | 2.5 | 0.6 | 0.2 | 1e-6 |
| **Llama3-Instruct-v0.2** | 10.0 | 0.4 | 0.2 | 1e-6 |
| **Gemma2-Instruct** | 10.0 | 0.4 | 5e-2 | 8e-7 |

**Hyperparameter in $\alpha$-DPO.** Table 4 outlines the hyperparameters used for $\alpha$-DPO under various settings. It's important to note that while our approach involves three key parameters, we have found through experience that $\beta$ can be reliably set to 10.0 by default. Among these parameters, $\gamma$ typically requires more careful tuning. As for $\alpha$, we have observed consistent performance improvements when set to 5e-2 by default. If you are already familiar with the parameter settings for SimPO, you can focus your search primarily on $\alpha$ or simply adopt the default setting of $\alpha = 5e - 2$.

**Decoding hyperparameters.** The decoding hyperparameters used in this study are the same as those employed by SimPO[4]. We extend our sincere gratitude to the SimPO team for sharing their invaluable insights.

**Computation environment.** All training experiments presented in this paper were conducted using 8×A100 GPUs, as per the procedures detailed in the alignment-handbook repository.[5]

---

[4] https://github.com/princeton-nlp/SimPO/tree/main/eval
[5] https://github.com/huggingface/alignment-handbook

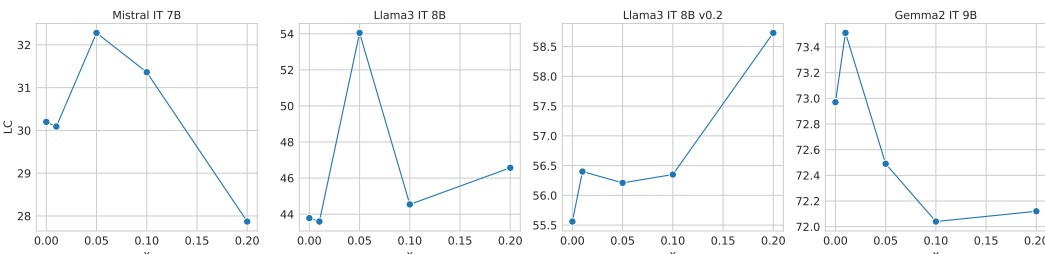

Figure 5: $\alpha$-DPO LC on AlpacaEval 2 with different $\alpha$ values.

## C.2 $\alpha$-DPO WITHOUT LENGTH-NORMALIZED

In this paper, we consider length-normalized training as a stability technique and not as a primary contribution of this work. Existing research (Meng et al., 2024) has demonstrated that length normalization can indeed enhance model performance, particularly with respect to the length control win rate. However, to validate the general applicability of $\alpha$-DPO—specifically, its stability and performance without length normalization—we conducted experiments across several models: meta-llama/Meta-Llama-3-8B-Instruct, mistralai/Mistral-7B-Instruct-v0.2, and google/gemma-2-9b-it.

We evaluated DPO, SimPO without length normalization, and $\alpha$-DPO without length normalization. The experimental results, as shown in Table 5, demonstrate that $\alpha$-DPO consistently achieves performance improvements even without the use of length normalization. This indicates the robustness and general effectiveness of $\alpha$-DPO.

Table 5: Performance comparison without length-normalization on AlpacaEval2. "LC" denotes the length-controlled win rate, and "WR" represents the raw win rate.

| Method | Llama3-Instruct (8B) | | Mistral-Instruct (7B) | | Llama3-Instruct v0.2 (8B) | | Gemma2-Instruct (9B) | |
|---|---|---|---|---|---|---|---|---|
| | LC (%) | WR (%) | LC (%) | WR (%) | LC (%) | WR (%) | LC (%) | WR (%) |
| DPO | 40.2 | 38.1 | 20.3 | 18.0 | 51.1 | 53.3 | 70.2 | 66.9 |
| SimPO w/o LN | 42.4 | 40.4 | 30.5 | 38.2 | 49.2 | 52.6 | 71.2 | 69.9 |
| $\alpha$-DPO w/o LN | **44.4** | **42.6** | **32.0** | **38.4** | **51.1** | **54.0** | **72.7** | **70.5** |

## C.3 $\alpha$-DPO WITH DIFFERENCT $\alpha$

To analyze the impact of $\alpha$ on the model, we adjust its value for four different models. The results are illustrated in Figure 5. When $\alpha$ is set to 0, the model degenerates to SimPO. As $\alpha$ increases, performance improves across all models, although the optimal value of $\alpha$ varies among them. This highlights the significance of $\alpha$.

It is noteworthy that within the parameter tuning range [1e-2, 5e-2, 0.1, 0.2], the optimal $\alpha$ values are consistently around 0.1 or even closer to 5e-2. This observation aligns with our Lemma 4.2, which indicates that smaller $\alpha$ values result in a lower estimation error in the online SimPO.

## C.4 COMPARISON WITH TDPO

To investigate the relationship between TDPO and $\alpha$-DPO, we conducted the experiments, with the results outlined below.

In its original form, TDPO did not perform well on LLAMA2-8B. By applying Lemma 4.3, we modified the expression $M(x, y_w, y_l)$ in $\alpha$-DPO to use TDPO's $\delta(x, y_w, y_l)$, converting our sentence-level estimations to a token-level calculation. This adjustment resulted in a noticeable performance improvement, which we attribute to the length-normalization, $\gamma$ and z-score normalization of $\delta(x, y_w, y_l)$. Nevertheless, the modified TDPO still underperformed compared to $\alpha$-DPO. This indicates that, when the $\pi_{\text{ref}}$ is suboptimal, token-level calculations are prone to significant errors.

Table 6: Performance comparison between TDPO and $\alpha$-DPO.

| Method | Llama3-Instruct (8B) | |
| --- | --- | --- |
| | LC (%) | WR (%) |
| TDPO | 52.8 | 45.9 |
| $\alpha$-DPO w/ $\delta(x, y_w, y_l)$ | 56.9 | 50.4 |
| $\alpha$-DPO w/ $M(x, y_w, y_l)$ | 58.7 | 51.1 |

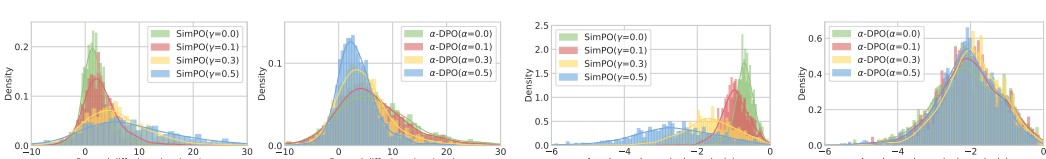

Figure 6: (a) SimPO: Reward difference distribution under different $\gamma$ values. (b) $\alpha$-DPO: Reward difference distribution under different $\alpha$ values. (c) SimPO: Log likelihood distribution on chosen responses under different $\gamma$ values. (d) $\alpha$-DPO: Log likelihood distribution on chosen responses under different $\alpha$ values.

