# OpenReview forum: "$\alpha$-DPO: Adaptive Reward Margin is What Direct Preference Optimization Needs"
_ICLR.cc/2025/Conference — Submitted to ICLR 2025_

### Official Review · Reviewer_8xwP · 2024-10-31

**Soundness:** 3
**Presentation:** 3
**Contribution:** 3
**Rating:** 5
**Confidence:** 3

**Summary:**

This paper extended the SimDPO to an adaptive setting where the reference model is a mixture of the uniform distribution and the original reference model $\pi_{\text{ref}}$. The authors provide some theoretical justification and experiments demonstrate the performance of the proposed algorithm.

**Strengths:**

- This paper is well-written and easy to follow, the author provide a good motivation to extend the SimPO to a mixture setting
- The authors tried to provide some justification building the connection between the SimPO and the proposed method
- Extensive experiments are conducted on verification

**Weaknesses:**

- It seems to me that this method is just a one-step extension to the SimPO to make the reference model a tunable mixture of the uniform distribution and original reference model. The authors might want to demonstrate the technical challenges more in order to make this extension trivially 'tuning the parameter in a super set'
- The theoretical justification in Definition 4.1 and later is not well supported. For example, in definition 4.1, the authors write the expectation as $(x, y_w, y_l) \sim \pi_{\text{old}}$. I assume that the authors are trying to say $x$ is from the prompt dataset and $y_w, y_l$ are *independently* sampled from the $\pi_{\text{old}}(\cdot | x)$, according to this definition. Even regardless of the flaw of this notation, there are still some issue in this definition: the role that $y_w$ and $y_l$ plays are not equal, so in online DPO, another oracle needs to be called to compare between $y_l$ and $y_w$ and generate the preference label. Also, an online DPO justification might not be convincing enough to justify the offline DPO methods.
- Following up withe the previous theoretical justification, the transit between eq(15) and eq(16) is not convincing. It seems that from Definition 4.1, eq (15) is well supported, but eq (16) is connected with the proposed algorithm, thus the authors directly draw the connection by flipping the $\pi_{\text{ref}}$ and $\pi_{\text{old}}$. I would suggest the authors to improve the justification for section 4, and reconsider the contribution of this part from over claimed.

**Questions:**

- It seems that when $\alpha = 0$, the algorithm becomes the SimDPO. I wonder when the algorithm becomes DPO?

---

> ### Author Response · Authors · 2024-11-21
>
> We thank the reviewer for their thoughtful feedback and constructive suggestions. Below, we address your comments in detail.
>
> ---
> **Q1: It seems to me that this method is just a one-step extension to the SimPO to make the reference model a tunable mixture of the uniform distribution and original reference model.**
>
> A1: While $\alpha$-DPO has a mixture form, we do not view this as a weakness. Recent work, such as [1], highlights the importance of the reference model and its impact on performance, with SimPO introducing the concept of being reference-free. Our work aims to explore the optimal reference model needed for offline preference learning without additional gold RM models. We hope to provide new insights, for example, by integrating online algorithms and KL Divergence control.
>
> [1] Liu et al. 2024. Understanding Reference Policies in Direct Preference Optimization. arXiv preprint arXiv:2407.13709
>
> **Q2: In online DPO, another oracle needs to be called to compare between $y\_w$ and $y\_l$ and generate the preference label.**
>
> A2:Thank you for your suggestion. We acknowledge the issue of inconsistency with the notation, and we will revise this section in the draft for clarity. The corrected description of the online process is as follows:
>
> Let's revisit the acquisition method for these samples (e.g., using princeton-nlp/llama3-ultrafeedback dataset):
> For each prompt $x$, we generate 5 responses using the SFT model with a sampling temperature of 0.8. We then score these responses with llm-blender/PairRM and select the highest-scoring one as $y\_w$ and the lowest-scoring one as $y\_l$.
> For the highest-scoring sample $y\_w$, it follows a new conditional distribution:
>
> $$\pi(y \mid x, \text{score}(y) = \max(\{\text{score}(y\_1), \ldots, \text{score}(y\_5)\}))\approx\pi(win|x),$$
> Where the last part uses the fact that for a given x the generative model has a uniform chance to generate all possible y.
> Similarly, for the lowest-scoring sample $y\_l$, it follows:
> $$\pi(y \mid x, \text{score}(y) = \min(\{\text{score}(y\_1), \ldots, \text{score}(y\_5)\}))\approx \pi(lose|x)$$
> We have ensured that the notation is now consistent and precise across the descriptions.
>
> **For online DPO, ideally, we would utilize an oracle reward model to evaluate each sampled instance. However, this approach incurs significant computational costs. Therefore, we aim to approximate this effect through importance sampling, as described in Equation 14.**
>
> **Q3: An online DPO justification might not be convincing enough to justify the offline DPO methods**
>
> A3:
> It is well known that preference optimization with online ingredients, particularly methods like online AI feedback (OAIF) [1] and self-play [2,3], enhances model alignment by generating new data during training.
> However, due to the computation cost sample regeneration and training stability issues, the pure online type preference optimization methods are not well applied in industrial practice.  Based on this observation, we thus proposed to study an interesting research topic, i.e., `can we mimic the online feature in the classic offline setting?` Our theoretical analysis is trying to close the gap with the importance sampling trick.
>
> In particular, the underlying logic here first defines the expression for online SimPO, which is characterized by the continuous updating of sampled data during training. In this process, the data transitions from the offline set $(y\_w, y\_l) \in D$ to the online set $(y\_w, y\_l) \in \pi\_\theta$. This operation can be viewed as an important sampling method. Interestingly, this key technique of importance sampling aligns with the optimization direction of $\alpha$-DPO loss, with both approaches converging in the scenario where $\alpha \to 0$. Consequently, Lemma 4.2 aims to establish a connection between $\alpha$-DPO and the Online SimPO loss, thereby facilitating the integration of these concepts.
>
> [1] Guo et al. Direct Language Model Alignment from Online AI Feedback. CoRR abs/2402.04792 (2024)
>
> [2] Chen et al. Self-Play Fine-Tuning Converts Weak Language Models to Strong Language Models. ICML 2024.
>
> [3] Wu et al. Self-Play Preference Optimization for Language Model Alignment. CoRR abs/2405.00675 (2024)

---

> ### Author Response · Authors · 2024-11-21
>
> **Q4: The transition between eq(15) and eq(16) is not convincing.**
>
> A4: To elucidate the rationale behind this transition, we offer the following explanations.
>
> 1. **Necessity of the Transformation:**
>
> When the condition $\pi\_{\theta\_{\text{old}}}(y\_l|x) > \pi\_{\text{ref}}(y\_l|x)$ arises—potentially due to the overrepresentation of less preferred responses in $\pi\_{\theta\_{\text{old}}}$—the importance weight $w(y\_w, y\_l | x)$ becomes disproportionately large. This disproportionate weight amplifies the impact of less preferred responses within the expectation, which could consequently result in suboptimal updates to $\pi\_\theta$. To mitigate this issue, an adjustment is made by placing $\pi\_{\theta\_{\text{old}}}(y\_l|x)$ in the denominator, thereby inversely scaling the weight associated with the overrepresented less preferred responses $y\_l$. This adjustment serves to diminish the undue influence of these responses that have higher probabilities under $\pi\_{\theta\_{\text{old}}}$ compared to $\pi\_{\text{ref}}$.
>
> 2. **Supporting the Transition Between Equations (15) and (16) Through Simulation of Online DPO Training Steps:**
>
> Consider the scenario at step $t$  of the optimization process, where we refine $(x,y\_w,y\_l)$. In this scenario, the updated model $\pi\_\theta^{t+1}$ assigns a higher score to $y\_w$ and a lower score to $y\_l$. Consequently, when new samples are generated using $\pi\_\theta^{t+1}$, the probability of generating $y\_w$ in response to the same prompt $x$ increases, while the probability of generating $y\_l$ decreases. This aligns with the corrected importance weight direction, given by:
> $$w\_{\text{corr}}(y\_w, y\_l|x) = \frac{\pi\_{\theta\_{\text{old}}}(y\_w|x)}{\pi\_{\text{ref}}(y\_w|x)} \cdot \frac{\pi\_{\text{ref}}(y\_l|x)}{\pi\_{\theta\_{\text{old}}}(y\_l|x)}$$
> We attribute the perceived inconsistency to a lack of rigorous symbol usage, which can be clarified by considering the following conditional distributions as mentioned in A1:
> - For the response with the highest score $y\_w$, it adheres to the conditional distribution:
> $$\pi(y \mid x, \text{score}(y) = \max(\{\text{score}(y\_1), \ldots, \text{score}(y\_5)\}))\approx\pi(win|x),$$
> - For the response with the lowest score $y\_l$, it adheres to the conditional distribution:
> $$\pi(y \mid x, \text{score}(y) = \min(\{\text{score}(y\_1), \ldots, \text{score}(y\_5)\}))\approx \pi(lose|x)$$

---

> ### Author Response · Authors · 2024-11-21
>
> 3. **A Theoretical Justification via Reward Difference Estimation**
>
> The original DPO objective maximizes the probability $p^*(y\_{\mathrm{win}} \succ y\_{\mathrm{lose}} \mid x)$, which, under the Bradley-Terry (BT) model, can be formulated as maximizing the following log-likelihood:
> $$\max \log \sigma(r\_\theta(x, y\_{\mathrm{win}}) - r\_\theta(x, y\_{\mathrm{lose}})),$$
> where $\sigma$ is the sigmoid function. Defining the reward difference $\delta = r\_\theta(x, y\_{\mathrm{win}}) - r\_\theta(x, y\_{\mathrm{lose}})$, the optimization objective simplifies to:
> $$\max \log \sigma(\delta) := \mathcal{T}(\delta).$$
> Suppose we start with an initial $\delta\_0$. A possible improved estimate $\delta\_1$ that maximizes $\mathcal{T}(\delta)$ is given by:
>
> $$\delta\_1 = \delta\_0 + \alpha \nabla \mathcal{T}(\delta\_0) = \delta\_0 + \alpha G(\sigma(-\delta\_0)),$$
> where $G$ is a monotonic function preserving the direction of $\sigma(-\delta\_0)$. This implies that for sufficient samll$\alpha$, $\delta\_1$yields  higher likelihood and can be viewed as a more optimistic estimation of the reward difference than $\delta\_0$.
> In the case of DPO, the reward difference $\delta$ can be reformulated as:
> $$\delta\_0 = \frac{\log \pi\_\theta(y\_{\mathrm{win}} \mid x)}{\log \pi\_{\text{ref}}(y\_{\mathrm{win}} \mid x)} - \frac{\log \pi\_\theta(y\_{\mathrm{lose}} \mid x)}{\log \pi\_{\text{ref}}(y\_{\mathrm{lose}} \mid x)}.$$
> Updating $\delta$ using $\delta\_1$ leads to:
>
> $\begin{align}
>     \delta\_1 &= \frac{\log \pi\_{\theta}(y\_{\mathrm{win}}|x)}{\log \pi\_{\mathrm{ref}}(y\_{\mathrm{win}}|x)} - \frac{\log \pi\_{\theta}(y\_{\mathrm{lose}}|x)}{\log \pi\_{\mathrm{ref}}(y\_{\mathrm{lose}}|x)} + \alpha G\left(\sigma\left(- \frac{\log \pi\_{\theta}(y\_{\mathrm{win}}|x)}{\log \pi\_{\mathrm{ref}}(y\_{\mathrm{win}}|x)} + \frac{\log \pi\_{\theta}(y\_{\mathrm{lose}}|x)}{\log \pi\_{\mathrm{ref}}(y\_{\mathrm{lose}}|x)}\right) \right)\notag \\\\
>     & = \frac{\log \pi\_{\theta}(y\_{\mathrm{win}}|x)}{\log \pi\_{\theta}(y\_{\mathrm{lose}}|x)} + \left(\gamma - \frac{\log \pi\_{\mathrm{ref}}(y\_{\mathrm{win}}|x)}{\log \pi\_{\mathrm{ref}}(y\_{\mathrm{lose}}|x)}\right) -\left[\gamma -\alpha G\left(\sigma\left(-\frac{\log \pi\_{\theta}(y\_{\mathrm{win}}|x)\log \pi\_{\mathrm{ref}}(y\_{\mathrm{lose}}|x)}{\log \pi\_{\mathrm{ref}}(y\_{\mathrm{win}}|x)\log \pi\_{\theta}(y\_{\mathrm{lose}}|x)}\right)\right)\right]
> \end{align}$
>
> If we take the assumption that $\gamma \approx \log \pi\_{\mathrm{ref}}(y\_{\mathrm{win}}|x) - \log \pi\_{\mathrm{ref}}(y\_{\mathrm{lose}}|x)$ holds per uniform assumption, then above equation implies
> $$\delta\_1 \approx \frac{\log \pi\_{\theta}(y\_{\mathrm{win}}|x)}{\log \pi\_{\theta}(y\_{\mathrm{lose}}|x)} -\left[\gamma -\alpha G\left(\sigma\left(-\frac{\log \pi\_{\theta}(y\_{\mathrm{win}}|x)\log \pi\_{\mathrm{ref}}(y\_{\mathrm{lose}}|x)}{\log \pi\_{\mathrm{ref}}(y\_{\mathrm{win}}|x)\log \pi\_{\theta}(y\_{\mathrm{lose}}|x)}\right)\right)\right]$$
> The remaining part follows by choosing$G(\cdot)\propto \sigma^{-1}(\cdot)$while maintaining monotonicity.
>
> In conclusion, our proposed method can be viewed as a new reward difference estimator (i.e.,$\delta\_1$ instead $\delta\_0$), which is more optimistic than its vanilla counterpart under the uniform assumption.
>
> **Q5: I wonder when the algorithm becomes DPO?**
>
> A5: Currently, the $\alpha$-DPO algorithm cannot be transformed into DPO merely through parameter adjustments, similar to how SimPO cannot be converted to DPO by altering $\gamma$. However, I believe this topic presents significant promise, allowing us to propose a more generalized formulation:
> $\hat{\pi}\_{\text{ref}}(\cdot|x)= U(\cdot|x) \pi\_\theta^{\alpha\_1}(\cdot|x) \pi\_{\text{ref}}^{\alpha\_2}(\cdot|x)$
> This expression encompasses several specific cases:
> - Setting $\alpha\_1=0$ and $\alpha\_2=1$ results in the DPO formulation with $\gamma$.
> - Setting $\alpha\_1=\alpha\_2=0$ recovers the SimPO expression.
> - Setting $\alpha\_1=\alpha, \alpha\_2=-\alpha$ yields the $\alpha$-DPO formulation.
>
> ---
> We hope these additional clarifications address your concerns comprehensively. Thank you again for your thoughtful review and the opportunity to improve our work.

---

> ### Author Response · Authors · 2024-12-04
> **We are grateful for the opportunity to refine our work through this constructive feedback process.**
>
> We greatly appreciate the opportunity to engage in a substantive dialogue with you, aimed at further improving the quality and clarity of our work. Below, we provide a detailed summary addressing your key concerns. We hope that this explanation will sufficiently address your questions and merit your endorsement.
>
> > ### 1. **The proposed reference model is a tunable mixture of the uniform distribution and orginal refernece model.**
> In our response [A1](https://openreview.net/forum?id=QqziJAdev9&noteId=jULBTfkf5a), we clarify that this formulation is neither a weakness nor merely a trivial extension achieved by "tuning the parameter in a superset."
>
> First, let us clarify that the primary motivation for proposing the reference policy in Eq. (8) arises from the limitations of the conventional $\pi\_{\text{ref}} = \pi\_{\text{SFT}}$ setup, which often exhibits significant randomness and errors. Addressing this limitation is a core objective of our work, a perspective that is similarly reflected in SimPO [1] (Section 3.2 and Figure 4b) and Liu et al. [2] (Section 5).
>
> Our goal is thus to design a more robust and effective reference policy, as expressed in Eq. (8). To provide additional context, we elaborate on the following key points:
>
> - **Alignment with online SimPO optimization.** Preference optimization with online elements (e.g., online AI feedback [3] and self-play [4][5]) is widely acknowledged to improve model alignment by generating new data during training. However, the high computational cost and instability of purely online methods limit their practicality in industrial applications. Our work explores whether online-like behavior can be emulated in a classical offline setting. Theoretical analysis based on importance sampling is provided to bridge this gap.
>
> - **Structural advantages of $r(x, y\_w) - r(x, y\_l) - \delta$.** A major contribution of token-level DPO is the introduction of the form $r(x, y\_w) - r(x, y\_l) - \delta$, which is similar to DPO with an offset and helps control the KL divergence. Our $\alpha$-DPO framework leverages the form $r(x, y\_w) - r(x, y\_l) - M$, which enhances performance by using $M$ instead of $\delta$. Appendix Table 6 provides empirical evidence, showing the performance advantages of $\alpha$-DPO over TDPO.
>
> - **Mitigating the impact of label flipping noise.** Within the SimPO framework, the gradient is expressed as:
>   $$
>   \nabla\_\theta \mathcal{L}\_{\mathrm{SimPO}}(\pi\_\theta) = -\beta \mathbb{E}\_{(x, y\_w, y\_l) \sim \mathcal{D}} \left[ s\_\theta \left( \frac{1}{|y\_w|} \nabla\_\theta \log \pi\_\theta(y\_w|x) - \frac{1}{|y\_l|} \nabla\_\theta \log \pi\_\theta(y\_l|x) \right) \right],
>   $$
>   where
>   $$
>   s\_\theta = \sigma \left( \frac{\beta}{|y\_l|} \log \pi\_\theta(y\_l|x) - \frac{\beta}{|y\_w|} \log \pi\_\theta(y\_w|x) + \gamma \right).
>   $$
>   This formulation amplifies weights when the reward estimate is inaccurate (e.g., $\log \pi\_\theta(y\_l|x) - \log \pi\_\theta(y\_w|x) > 0$), increasing noise in gradients. In contrast, $\alpha$-DPO introduces an additional term:
>   $$
>   s\_\theta = \sigma \left( \frac{\beta}{|y\_l|} \log \pi\_\theta(y\_l|x) - \frac{\beta}{|y\_w|} \log \pi\_\theta(y\_w|x) + \gamma + \alpha M(x, y\_w, y\_l) \right).
>   $$
>   This term increases the weight when the reward estimate is accurate and decreases it when the reward estimate is inaccurate, mitigating noise amplification.
>
> - **Improving reference policy quality.** This work directly addresses the unreliability of reference policies (Section 3.1). By integrating the policy model into the reference model's design, we enhance the quality of the reference model and improve fine-tuning performance. Similar ideas have been explored in recent works [2][6], highlighting the broader relevance of this approach.
>
> References:
> [1] Meng et al. (2024): Simpo: Simple preference optimization with a reference-free reward. NeurIPS 2024.
> [2] Liu et al. (2024): Understanding Reference Policies in Direct Preference Optimization. arXiv preprint arXiv:2407.13709.
> [3] Guo et al. (2024): Direct Language Model Alignment from Online AI Feedback. CoRR abs/2402.04792.
> [4] Chen et al. (2024): Self-Play Fine-Tuning Converts Weak Language Models to Strong Language Models. ICML 2024.
> [5] Wu et al. (2024): Self-Play Preference Optimization for Language Model Alignment. CoRR abs/2405.00675.
> [6] Gorbatovski et al. (2024): Learn your reference model for real good alignment. arXiv preprint arXiv:2404.09656.

---

> ### Author Response · Authors · 2024-12-04
> **We are grateful for the opportunity to refine our work through this constructive feedback process.**
>
> > ### 2. **The significance of theoretical guarantees**
>
> In our response [A2](https://openreview.net/forum?id=QqziJAdev9&noteId=jULBTfkf5a), we revisited the distributions of $y\_w$ and $y\_l$ and revised the original text to ensure consistency throughout.
>
> In our response [A3](https://openreview.net/forum?id=QqziJAdev9&noteId=jULBTfkf5a), we explained the motivation for establishing connections with online algorithms. The core rationale is that online algorithms remain the most effective choices due to their superior performance. However, purely online preference optimization methods are often impractical for industrial applications due to computational cost, sample regeneration, and training stability issues. As a compromise, importance sampling is frequently employed, and similar approaches are widely adopted in the reinforcement learning literature [1][2][3][4].
>
> Finally, we would like to emphasize that SimPO has already demonstrated its effectiveness in various experimental settings. Our method extends SimPO to personalized scenarios, with the parameter $\alpha$ acting as a key control factor. Specifically:
> - The choice of $\alpha$ balances two competing objectives. When $\alpha = 0$, the method reduces to SimPO, where all training samples are assigned the same target margin.
> - As $\alpha$ increases, the model places greater emphasis on samples with accurate reward estimates.
> - However, excessively large values of $\alpha$ cause the model to focus solely on a few samples with the most accurate reward estimates, neglecting the contributions of other samples, which can hinder the training process.
>
> This trade-off highlights the importance of selecting an appropriate $\alpha$ to achieve a balance between conservativeness and aggressiveness.
>
> [1] Sergey Levine et al.: "Offline Reinforcement Learning: Tutorial, Review, and Perspectives on Open Problems," CoRR abs/2005.01643 (2020).
> [2] Alberto Maria Metelli et al.: "Policy Optimization via Importance Sampling," NeurIPS 2018.
> [3] Tengyang Xie et al.: "Towards Optimal Off-Policy Evaluation for Reinforcement Learning with Marginalized Importance Sampling," NeurIPS 2019.
> [4] Philip J. Ball et al.: "Efficient Online Reinforcement Learning with Offline Data," ICML 2023.
>
> > ### 3. **The transition between eq(15) and eq(16) is not convincing.**
>
> In our response [A4](https://openreview.net/forum?id=QqziJAdev9&noteId=cRGp9hi2Tf), we provided a detailed analysis of the necessity and rationale behind this transition from multiple perspectives.
>
> ---
>
> Your constructive feedback has been invaluable, and we are committed to leveraging this discussion to improve our work. Thank you again for the opportunity to address your questions and for considering our responses.

---

### Official Review · Reviewer_EUrj · 2024-11-01

**Soundness:** 4
**Presentation:** 4
**Contribution:** 3
**Rating:** 6
**Confidence:** 4

**Summary:**

DPO and SimPO are two popular offline methods for LLM alignment. The authors demonstrate through theoretical analysis that SimPO is a special case of DPO where the reference model is assumed to be a uniform policy. As a result, DPO does not use an optimal reference policy while SimPO does not take into account data specific variances in defining a target margin. The authors address these limitations by proposing a novel loss function ($\alpha-DPO$) that relies on data specific differences to compute a dynamic reward margin. They theoretically demonstrate  ($\alpha-DPO$) is the lower bound of SimPO. They demonstrate through empirical results that the performance of ($\alpha-DPO$) is better than the baselines on multiple alignment benchmarks

**Strengths:**

1. The paper is easy to understand. The theoretical analysis showing the limitation of DPO and SimPo along with the proposed improvements seems logical
2. The empirical results show that this method outperforms DPO and SimPO with minimal additional complexity
3. The ablation studies are very useful in understanding the contributions of different changes to the loss functions

**Weaknesses:**

1. As discussed in the limitations, this requires tuning an additional parameter $\alpha$. It is not clear if a single $\alpha$ value is used for each pair of benchmark and model. If LC and raw WR values comes from different $\alpha$ values then the results are slightly misleading since it's not just one model being used for comparison against the benchmarks.

**Questions:**

N/A

---

> ### Author Response · Authors · 2024-11-21
>
> We thank the reviewer for their thoughtful feedback and constructive suggestions. Below, we address your comments in detail.
>
> ---
> **Q1: If LC and raw WR values come from different α values, then the results are slightly misleading since it's not just one model being used for comparison against the benchmarks.**
>
> A1: Thank you for your insightful suggestion. `We would like to clarify that all LC and raw WR values reported in our experiments were obtained using a fixed $\alpha$ value across all benchmarks.`
>
> We observed that introducing $\alpha$-DPO leads to performance improvements across all benchmarks, which maintains the state-of-the-art performance, albeit with varying degrees of relative improvement. We attribute this observational phenomenon to the current limitations in benchmark evaluation methods, possibly due to the inconsistencies introduced by GPT-4's judgment as a metric.
>
> This trend is not unique to $\alpha$-DPO; it also occurs with other methods, meaning each method has different rankings depending on the benchmark. We believe that developing a truly rigorous and effective benchmark represents an intriguing future research direction.
>
> ---
> We hope these additional clarifications address your concerns comprehensively. Thank you again for your thoughtful review and the opportunity to improve our work.

---

### Official Review · Reviewer_RjJZ · 2024-11-04

**Soundness:** 1
**Presentation:** 3
**Contribution:** 2
**Rating:** 3
**Confidence:** 5

**Summary:**

This paper studies how to align LLMs with human preferences. The authors propose a new algorithm $\alpha$-DPO which adaptively set the reward margin based on the ratio between the preference model and the policy model. They prove that the objective of $\alpha$-DPO is a lower bound on the online SimPO loss. They also conduct experiments on AlpacaEval 2 and Arena-Hard to validate the empirical performance of $\alpha$-DPO.

**Strengths:**

a. The authors propose a new objective for LLM alignment and conduct extensive experiments on several benchmarks. The proposed algorithm outperforms baselines on these benchmarks.

b. The presentation is clear and easy to follow.

**Weaknesses:**

a. The proposed method is not technically sound. The derivation begins with an implicit reference model, but this model is neither well-motivated nor justified. First, Equation 8 does not have a normalization factor, and tuning the hyperparameter $\alpha$ very likely results in an invalid distribution. Additionally, it’s unclear why this implicit reference model is necessary instead of using a standard SFT model. The authors consider a special case with $\alpha=1$, incorporating the ratio between the policy model and the reference model, but there is no clear rationale for why such a ratio is required.

b. The theoretical analysis is problematic. First, relating the objective to the online SimPO loss is not meaningful, as the online SimPO loss itself lacks theoretical guarantees. Second, Lemma 4.2 claims that the objective of $\alpha$-DPO provides a lower bound on the online SimPO loss. However, minimizing a lower bound is questionable since the gap between the true value and the lower bound is unknown. Minimizing an upper bound would be more meaningful. The statement, “the lower-bounding property provides theoretical guarantees that … not perform worse than online SimPO loss, ensuring convergence to a well-generalized policy,” is confusing. How can minimizing a lower bound provide such a strong theoretical guarantee?

c. The experimental improvement is marginal, typically less than 1.5%. Given that benchmark evaluations such as AlpacaEval 2 and Arena-Hard rely on GPT-4’s judgment, which can vary by 1-2%, these improvements may not be convincing, especially when the method involves at least one more hyperparameter than the baselines. Furthermore, the authors evaluate performance on only two benchmarks, which is limited, particularly in LLM alignment experiments. More evaluations on academic benchmarks like MT-Bench, MMLU, GSM8K, and TruthfulQA are required. Additionally, most experiments involve models already trained using RLHF methods. Testing on models without RLHF, such as Llama-3-8B, would be necessary to confirm that the proposed algorithm does not rely on pre-existing alignment.

**Questions:**

See Weaknesses.

---

> ### Author Response · Authors · 2024-11-21
>
> We thank the reviewer for their thoughtful feedback and constructive suggestions. Below, we address your comments in detail.
>
> ---
> **Q1: First, Equation 8 does not have a normalization factor, and tuning the hyperparameter $\alpha$ very likely results in an invalid distribution.**
>
> A1: We recognize that the expression lacks rigor. A more precise formulation is:
>
> $$\hat{\pi}\_{\text{ref}}(y|x) \propto U(y|x)\left(\frac{\pi\_\theta(y|x)}{\pi\_{\text{ref}}(y|x)}\right)^\alpha.$$
> This expression combines the benefits of DPO and SimPO. We have revised the manuscript to reflect this clarification.
>
> **Q2: Additionally, it’s unclear why this implicit reference model is necessary instead of using a standard SFT model.**
>
> A2: The assumption that the SFT has large errors serves as the motivation for proposing $\alpha$-DPO. This assumption is discussed in Section 3.1, where we highlight the unreliability of the reference policy in DPO.
>
> **Q3: The authors consider a special case with $\alpha=1$, incorporating the ratio between the policy model and the reference model, but there is no clear rationale for why such a ratio is required.**
>
> A3: The motivation for the proposed reference policy $\hat{\pi}\_{\text{ref}}(y|x)$ can be clarified as follows:
> 1. **Utility Theory Perspective:**
> The proposed $\hat{\pi}\_{\text{ref}}(y|x)$ is designed with the uniform distribution $U(y|x)$ as a baseline. The term $\left(\frac{\pi\_\theta(y|x)}{\pi\_{\text{ref}}(y|x)}\right)^\alpha$ dynamically adjusts the reward margin by balancing contributions from the policy and reference models. This mechanism can be interpreted through the lens of utility theory as relative attractiveness, enabling adaptive instance-specific reward modeling.
> 2. **Gradient Perspective:**
> By introducing $\hat{\pi}\_{\text{ref}}(y|x)$, the framework mitigates the label flipping issues found in DPO or SimPO.
> In the SimPO framework, the gradient is expressed as:
> $$\nabla\_\theta\mathcal{L}\_{\mathrm{SimPO}}(\pi\_\theta)=-\beta\mathbb{E}\_{(x,y\_w,y\_l)\sim\mathcal{D}}\left[s\_\theta\left(\frac1{|y\_w|}\nabla\_\theta\log\pi\_\theta(y\_w|x)-\frac1{|y\_l|}\nabla\_\theta\log\pi\_\theta(y\_l|x)\right)\right],$$
>    where
> $s\_\theta=\sigma\left(\frac\beta{|y\_l|}\log\pi\_\theta(y\_l|x)-\frac\beta{|y\_w|}\log\pi\_\theta(y\_w|x)+\gamma\right)$.
>    This formulation may amplify weights when the reward estimate is incorrect. By contrast, under $\alpha$-DPO:
> $$s\_\theta=\sigma\left(\frac\beta{|y\_l|}\log\pi\_\theta(y\_l|x)-\frac\beta{|y\_w|}\log\pi\_\theta(y\_w|x)+\gamma + \alpha M(x,y\_w,y\_l)\right),$$
>    the additional $\alpha M(x,y\_w,y\_l)$ component increases weight when the reward estimate is accurate, ensuring a more robust reward signal.
>
> 3. **Motivational Core:**
> The central goal of the proposed $\alpha$-DPO is to address the unreliability of the reference policy, as outlined in Section 3.1. By integrating the policy model into the reference model design, the quality of the reference model is enhanced, improving fine-tuning performance. Similar concepts have been explored in recent works [1][2]:
>
> `We have also provided a theoretical justification through reward difference estimation. For more details, please refer to our response (A4) to Reviewer 8xwP.`
>
> [1] Gorbatovski et al. (2024): Learn your reference model for real good alignment. arXiv preprint arXiv:2404.09656.
> [2] Liu et al. (2024): Understanding Reference Policies in Direct Preference Optimization. arXiv preprint arXiv:2407.13709.

---

> ### Author Response · Authors · 2024-11-21
>
> **Q4: First, relating the objective to the online SimPO loss is not meaningful, as the online SimPO loss itself lacks theoretical guarantees.**
>
> A4: It is well known that preference optimization with online ingredients, particularly methods like online AI feedback (OAIF) [1] and self-play [2,3], enhances model alignment by generating new data during training.
> However, due to the computation cost sample regeneration and training stability issues, the pure online type preference optimization methods are not well applied in industrial practice.  Based on this observation, we thus proposed to study an interesting research topic, i.e., `can we mimic the online feature in the classic offline setting?` Our theoretical analysis is trying to close the gap with the importance sampling trick.
>
> In particular, the underlying logic here first defines the expression for online SimPO, which is characterized by the continuous updating of sampled data during training. In this process, the data transitions from the offline set $(y\_w, y\_l) \in D$ to the online set $(y\_w, y\_l) \in \pi\_\theta$. This operation can be viewed as an important sampling method. Interestingly, this key technique of importance sampling aligns with the optimization direction of $\alpha$-DPO loss, with both approaches converging in the scenario where $\alpha \to 0$. Consequently, Lemma 4.2 aims to establish a connection between $\alpha$-DPO and the Online SimPO loss, thereby facilitating the integration of these concepts.
>
> [1] Guo et al. Direct Language Model Alignment from Online AI Feedback. CoRR abs/2402.04792 (2024)
>
> [2] Chen et al. Self-Play Fine-Tuning Converts Weak Language Models to Strong Language Models. ICML 2024.
>
> [3] Wu et al. Self-Play Preference Optimization for Language Model Alignment. CoRR abs/2405.00675 (2024)
>
> **Q5: However, minimizing a lower bound is questionable since the gap between the true value and the lower bound is unknown. Minimizing an upper bound would be more meaningful.**
>
> A5: We apologize for any confusion. A more precise statement is needed:
> For any policy model $\pi\_\theta$ and reference model $\pi\_{\text{ref}}$, there exists a sufficiently small $\alpha$ such that the following inequalities hold:
> $$|\mathcal{L}\_{\text{SimPO}}^{\text{online}}(\pi\_\theta, \pi\_{\text{ref}}) -\mathcal{L}\_{\text{$\alpha$-DPO}}(\pi\_\theta, \pi\_{\text{ref}})|\leq  \varepsilon(\alpha),$$
> where $\varepsilon(\alpha) = \mathbb{E}\_{\pi\_{\text{ref}}}\left[ \alpha |B| \left| \log \sigma(A) - \sigma(A) + 1 \right| \right].$
>
> This result demonstrates that optimizing $\alpha$-DPO provides a tight approximation of online-SimPO, which is crucial for achieving performance improvements. In response to feedback, we have revised Lemma 4.2 and included the supporting proofs. Thank you for your valuable suggestion.
>
> **Q6: The experimental improvement is marginal, typically less than 1.5%.**
>
> A6: We believe $\alpha$-DPO's improvements are significant. For example, AlpacaEval2 (LC) shows improvements near 6% on most models. Arena-Hard benchmarks also demonstrate over 5% gains, except for gemma2-9b.
> We conduct a comparison of the relevant results from Table 1, focusing on SimPO and $\alpha$-DPO.
> Detailed results are provided:
>
> | Metric        | LC | WR | SC | LC | WR | LC | WR | SC | LC | WR |
> |--------------|--------|--------|--------|--------|--------|--------|--------|--------|--------|--------|
> | SimPO        | 43.8 | 38.0 | 33.5 | 33.5 | 32.6 | 30.2 | 32.1 | 25.6 | 19.8 | 20.1 |
> | $\alpha$-DPO | 46.6 | 38.1 | 34.1 | 34.2 | 33.3 | 32.3 | 32.6 | 27.2 | 21.5 | 21.5 |
> | improv.      | 6.39% | 0.26% | 1.79% | 2.09% | 2.15% | 6.95% | 1.56% | 6.25% | 8.59% | 6.97% |
>
> | Metric        | LC | WR | SC | LC | WR | LC | WR | SC | LC | WR |
> |---------|----------|----------|----------|-------|-------|-------|-------|-------|-------|-------|
> | SimPO   | 55.6     | 49.6     | 28.5     | 34.0  | 33.6  | 72.4  | 65.0  | 45.0  | 56.1  | 57.8  |
> | Method  | 58.7     | 51.1     | 30.8     | 36.3  | 35.7  | 73.4  | 66.1  | 48.6  | 59.3  | 60.8  |
> | improv. | 5.58%    | 3.02%    | 8.07%    | 6.76% | 6.25% | 1.38% | 1.69% | 8.00% | 5.70% | 5.19% |

---

> ### Author Response · Authors · 2024-11-21
>
> **Q7: Furthermore, the authors evaluate performance on only two benchmarks, which is limited, particularly in LLM alignment experiments. More evaluations on academic benchmarks like MT-Bench, MMLU, GSM8K, and TruthfulQA are required. Testing on models without RLHF, such as Llama-3-8B, would be necessary to confirm that the proposed algorithm does not rely on pre-existing alignment.**
>
> A7: Thank you for your suggestion. We have incorporated a new benchmark into our study. Please note that all experiments have been conducted using the Llama-3-8B-BASE model to address your concerns regarding pre-existing alignment.
>
> | Method                          | DPO (Llama3-8B-Base)  |SimPO (Llama3-8B-Base) | $\alpha$-DPO (Llama3-8B-Base)   |
> |---------------------------------|-------------------------|--------------|--------------|
> |truthfulqa\_mc2   | 53.66| 60.03| 62.89|
> |gsm8k  |  52.90  |  52.84  |53.90|
> | mmlu    |  62.14  |   62.05|62.43|
> ||||
> | MT-Bench   | 6.5| 6.6  |6.9|
> ||||
> | LC(AlpacaEval2)   | 14.92| 17.97  |22.69|
> | WR(AlpacaEval2)   | 13.02| 15.60  |20.47|
>
> These results confirm stable performance improvements, and the additional experiments will be included in the paper.
>
> ---
>
> We hope these additional clarifications address your concerns comprehensively. Thank you again for your thoughtful review and the opportunity to improve our work.

---

> > ### Comment · Reviewer_RjJZ · 2024-12-03
> >
> > Thanks for the detailed response. After reading it, I still have the following questions and concerns.
> >
> > 1. From the authors’ response, it seems the advantage of the new reference policy increases the weight of the gradient when the reward signal is correct. However, I am unclear why this ensures a more robust reward signal. Would it decrease the weight when the reward signal is incorrect? Additionally, there are no experimental results demonstrating that $\alpha$-DPO achieves better performance when the reward signal contains noise. If the advantage is simply to increase the weight, why not just use a larger $\beta$? The motivation for introducing $\hat \pi_{\text{ref}}$ is still unclear to me.
> >
> > 2. The theoretical guarantees of online algorithms largely rely on iteratively updating the preference dataset. Proving that the loss value of online SimPO and $\alpha$-DPO is close for a fixed dataset does not seem particularly meaningful. Furthermore, a close loss value does not imply much—at the very least, you need to show that the gradients are close. Moreover, is there any theoretical guarantee provided for SimPO?
> >
> > 3. Regarding the experimental results, my concern is that the win rate is judged by GPT-4, where the absolute judgment noise can vary by 1-2%. This makes the reported improvement appear marginal. Since the win rate is typically small, reporting relative improvement feels more like a way to inflate the percentage, which I find not very meaningful.

---

> > > ### Author Response · Authors · 2024-12-03
> > >
> > > **Q8: However, I am unclear why this ensures a more robust reward signal. Would it decrease the weight when the reward signal is incorrect?**
> > >
> > > **A8:** Yes, when the reward signal is incorrect (e.g., $ M(x, y\_w, y\_l) < 0 $), it decreases the weight as defined by
> > > $$ s\_\theta = \sigma\left(\frac{\beta}{|y\_l|}\log\pi\_\theta(y\_l|x) - \frac{\beta}{|y\_w|}\log\pi\_\theta(y\_w|x) + \gamma + \alpha M(x, y\_w, y\_l)\right). $$
> > >
> > >
> > > **Q9: Additionally, there are no experimental results demonstrating that $\alpha$-DPO achieves better performance when the reward signal contains noise.**
> > >
> > > **A9:** The existence of noise in reward signals has been well established in the literature, and addressing this issue is the primary motivation for methods such as cDPO, rDPO and SimPO. Our work assumes this general observation and focuses on demonstrating improvements under standard settings.
> > >
> > >
> > > **Q10: If the advantage is simply to increase the weight, why not just use a larger $\beta$? The motivation for introducing $\hat{\pi}\_{\text{ref}}$ is still unclear to me.**
> > >
> > >
> > > **A10:** First, the core motivation for introducing $\hat{\pi}\_{\text{ref}}$ is to address the unreliability of existing reference policies (consistent with the motivation of SimPO). Additionally, we aim to alleviate the limitations of SimPO’s fixed $\gamma$ setting for all samples. To this end, we propose a novel design for $\hat{\pi}\_{\text{ref}}$.
> > >
> > > Second, we have shown that $\hat{\pi}\_{\text{ref}}$ can be interpreted as a reweighting mechanism. Furthermore, we believe it can be analogized to a dynamic $\beta$ approach. However, since no existing work has implemented such a technique, we consider $\hat{\pi}\_{\text{ref}}$ to be a non-trivial contribution.
> > >
> > > Finally, we do not claim that the advantage of $\alpha$-DPO is simply to increase the weight. This is just one intuitive explanation among others. For example, additional advantages include:
> > >
> > > 1. **Addressing the unreliability of the reference policy (Section 3.1):** By integrating the policy model into the reference model, the quality of the reference model is improved, leading to enhanced fine-tuning performance. Similar concepts have been explored in recent studies [1][2].
> > >
> > > 2. **Balancing alignment and diversity via KL divergence control (Section 4.1):** We demonstrated that $\alpha$-DPO shares the same advantage as TDPO, effectively achieving KL divergence control.
> > >
> > >
> > > [1] Gorbatovski et al. (2024): Learn your reference model for real good alignment. arXiv preprint arXiv:2404.09656.
> > > [2] Liu et al. (2024): Understanding Reference Policies in Direct Preference Optimization. arXiv preprint arXiv:2407.13709.
> > >
> > > **Q11: Proving that the loss value of online SimPO and $\alpha$-DPO is close for a fixed dataset does not seem particularly meaningful.**
> > >
> > > **A11:** We respectfully disagree with the assertion that simulating online algorithms on a fixed dataset is not meaningful. Due to computational cost, sample regeneration, and training stability issues, purely online preference optimization methods are often not practical for industrial applications. A common alternative is to employ importance sampling. Similar approaches are prevalent in reinforcement learning literature [3][4][5][6].
> > >
> > > [3] Sergey Levine et al.: "Offline Reinforcement Learning: Tutorial, Review, and Perspectives on Open Problems," CoRR abs/2005.01643 (2020).
> > > [4] Alberto Maria Metelli et al.: "Policy Optimization via Importance Sampling," NeurIPS 2018.
> > > [5] Tengyang Xie et al.: "Towards Optimal Off-Policy Evaluation for Reinforcement Learning with Marginalized Importance Sampling," NeurIPS 2019.
> > > [6] Philip J. Ball et al.: "Efficient Online Reinforcement Learning with Offline Data," ICML 2023.

---

> > > ### Author Response · Authors · 2024-12-03
> > >
> > > **Q12: Furthermore, a close loss value does not imply much—at the very least, you need to show that the gradients are close.**
> > >
> > > **A12:** The gradient directions of online SimPO and $\alpha$-DPO are consistent. At optimization step $t$, refining $(x, y\_w, y\_l)$ updates $\pi\_\theta^{t+1}$, assigning higher probabilities to $y\_w$ and lower probabilities to $y\_l$. This adjustment ensures that newly generated samples with $\pi\_\theta^{t+1}$ align with the corrected importance weight direction. Below, we explicitly compare the gradients of the two methods:
> > >
> > > For **Online SimPO**, the gradient is given by:
> > > $$
> > > \nabla\_\theta\mathcal{L}\_{\mathrm{Online-SimPO}}(\pi\_\theta) = -\beta \mathbb{E}\_{(x, y\_w, y\_l) \sim \mathcal{D}} \left[ s\_\theta \left( \frac{1}{|y\_w|} \nabla\_\theta \log\pi\_\theta(y\_w|x) - \frac{1}{|y\_l|} \nabla\_\theta \log\pi\_\theta(y\_l|x) \right) \right],
> > > $$
> > > where
> > > $$
> > > s\_\theta = \frac{\pi\_{\theta\_{\text{old}}}(y\_w|x)}{\pi\_{\text{ref}}(y\_w|x)} \cdot \frac{\pi\_{\text{ref}}(y\_l|x)}{\pi\_{\theta\_{\text{old}}}(y\_l|x)} \cdot \sigma\left( \frac{\beta}{|y\_l|} \log\pi\_\theta(y\_l|x) - \frac{\beta}{|y\_w|} \log\pi\_\theta(y\_w|x) + \gamma \right).
> > > $$
> > >
> > > For **$\alpha$-DPO**, the gradient is:
> > > $$
> > > \nabla\_\theta\mathcal{L}\_{\alpha-\mathrm{DPO}}(\pi\_\theta) = -\beta \mathbb{E}\_{(x, y\_w, y\_l) \sim \mathcal{D}} \left[ d\_\theta \left( \frac{1}{|y\_w|} \nabla\_\theta \log\pi\_\theta(y\_w|x) - \frac{1}{|y\_l|} \nabla\_\theta \log\pi\_\theta(y\_l|x) \right) \right],
> > > $$
> > > where
> > > $$
> > > d\_\theta = \sigma\left( \frac{\beta}{|y\_l|} \log\pi\_\theta(y\_l|x) - \frac{\beta}{|y\_w|} \log\pi\_\theta(y\_w|x) + \gamma + \alpha M(x, y\_w, y\_l) \right).
> > > $$
> > >
> > > The primary difference lies in the importance weights ($s\_\theta$ for Online SimPO and $d\_\theta$ for $\alpha$-DPO). While Online SimPO adjusts weights based solely on the importance sampling correction term, $\alpha$-DPO incorporates the additional $\alpha M(x, y\_w, y\_l)$ term to further modulate the reward signal. This allows $\alpha$-DPO to refine alignment more effectively under scenarios where reward noise exists.
> > >
> > > Thus, both methods adjust the importance weights during training, prioritizing positive samples ($y\_w$) and penalizing negative samples ($y\_l$). However, the gradient modulation introduced by $\alpha M(x, y\_w, y\_l)$ in $\alpha$-DPO provides a more robust mechanism for alignment and diversity control.
> > >
> > > **Q13: Moreover, is there any theoretical guarantee provided for SimPO?**
> > >
> > > **A13:** In Section 3.1, we demonstrate that SimPO can be interpreted as DPO with a uniform reference distribution. This aligns with SimPO’s motivation to address the unreliability of the reference model by adopting a reference-free approach. Furthermore, SimPO has shown to be the most effective variant of DPO, justifying further exploration of its validity.
> > >
> > >
> > > **Q14: Regarding the experimental results, my concern is that the win rate is judged by GPT-4, where the absolute judgment noise can vary by 1–2%.**
> > >
> > > **A14:** We respectfully disagree with this concern. Benchmarks such as **AlpacaEval2 and ArenaHard** are widely considered the most reliable for evaluating LLM performance. Their adoption is largely due to the high consistency of GPT-4 judgments with human evaluations, **exceeding 98% agreement**.
> > >
> > > For instance, on AlpacaEval2 (LC metric), $\alpha$-DPO demonstrates significant improvements:
> > >
> > > - On Llama3-Instruct v0.2 (8B): 51.9 (DPO, NeurIPS 2023) → 55.6 (SimPO, NeurIPS 2024) → 58.7 ($\alpha$-DPO).
> > > - On Llama3-Instruct (8B): 40.2 (DPO, NeurIPS 2023) → 40.3 (R-DPO, ACL 2024) → 43.8 (SimPO, NeurIPS 2024) → 46.6 ($\alpha$-DPO).
> > >
> > > We believe these improvements are substantial and meaningful.

---

> > > > ### Author Response · Authors · 2024-12-04
> > > > **We are grateful for the insightful discussions and the opportunity to refine our work through this constructive feedback process.**
> > > >
> > > > We greatly appreciate the opportunity to engage in a substantive dialogue with you, aimed at further improving the quality and clarity of our work. Below, we provide a detailed summary addressing your key concerns. We hope that this explanation will sufficiently address your questions and merit your endorsement.
> > > >
> > > > > ### 1. **Motivation for the proposed reference policy (Eq. 8)**
> > > >
> > > > First, let us clarify that the primary motivation for proposing the reference policy in Eq. (8) arises from the limitations of the conventional $\pi\_{\text{ref}} = \pi\_{\text{SFT}}$ setup, which often exhibits significant randomness and errors. Addressing this limitation is a core objective of our work, a perspective that is similarly reflected in SimPO [1] (Section 3.2 and Figure 4b) and Liu et al. [2] (Section 5).
> > > >
> > > > Our goal is thus to design a more robust and effective reference policy, as expressed in Eq. (8). To provide additional context, we elaborate on the following key points:
> > > >
> > > > - **Alignment with online SimPO optimization.** Preference optimization with online elements (e.g., online AI feedback [3] and self-play [4][5]) is widely acknowledged to improve model alignment by generating new data during training. However, the high computational cost and instability of purely online methods limit their practicality in industrial applications. Our work explores whether online-like behavior can be emulated in a classical offline setting. Theoretical analysis based on importance sampling is provided to bridge this gap.
> > > >
> > > > - **Structural advantages of $r(x, y\_w) - r(x, y\_l) - \delta$.** A major contribution of token-level DPO is the introduction of the form $r(x, y\_w) - r(x, y\_l) - \delta$, which is similar to DPO with an offset and helps control the KL divergence. Our $\alpha$-DPO framework leverages the form $r(x, y\_w) - r(x, y\_l) - M$, which enhances performance by using $M$ instead of $\delta$. Appendix Table 6 provides empirical evidence, showing the performance advantages of $\alpha$-DPO over TDPO.
> > > >
> > > > - **Mitigating the impact of label flipping noise.** Within the SimPO framework, the gradient is expressed as:
> > > >   $$
> > > >   \nabla\_\theta \mathcal{L}\_{\mathrm{SimPO}}(\pi\_\theta) = -\beta \mathbb{E}\_{(x, y\_w, y\_l) \sim \mathcal{D}} \left[ s\_\theta \left( \frac{1}{|y\_w|} \nabla\_\theta \log \pi\_\theta(y\_w|x) - \frac{1}{|y\_l|} \nabla\_\theta \log \pi\_\theta(y\_l|x) \right) \right],
> > > >   $$
> > > >   where
> > > >   $$
> > > >   s\_\theta = \sigma \left( \frac{\beta}{|y\_l|} \log \pi\_\theta(y\_l|x) - \frac{\beta}{|y\_w|} \log \pi\_\theta(y\_w|x) + \gamma \right).
> > > >   $$
> > > >   This formulation amplifies weights when the reward estimate is inaccurate (e.g., $\log \pi\_\theta(y\_l|x) - \log \pi\_\theta(y\_w|x) > 0$), increasing noise in gradients. In contrast, $\alpha$-DPO introduces an additional term:
> > > >   $$
> > > >   s\_\theta = \sigma \left( \frac{\beta}{|y\_l|} \log \pi\_\theta(y\_l|x) - \frac{\beta}{|y\_w|} \log \pi\_\theta(y\_w|x) + \gamma + \alpha M(x, y\_w, y\_l) \right).
> > > >   $$
> > > >   This term increases the weight when the reward estimate is accurate and decreases it when the reward estimate is inaccurate, mitigating noise amplification.
> > > >
> > > > - **Improving reference policy quality.** This work directly addresses the unreliability of reference policies (Section 3.1). By integrating the policy model into the reference model's design, we enhance the quality of the reference model and improve fine-tuning performance. Similar ideas have been explored in recent works [2][6], highlighting the broader relevance of this approach.
> > > >
> > > > References:
> > > > [1] Meng et al. (2024): Simpo: Simple preference optimization with a reference-free reward. NeurIPS 2024.
> > > > [2] Liu et al. (2024): Understanding Reference Policies in Direct Preference Optimization. arXiv preprint arXiv:2407.13709.
> > > > [3] Guo et al. (2024): Direct Language Model Alignment from Online AI Feedback. CoRR abs/2402.04792.
> > > > [4] Chen et al. (2024): Self-Play Fine-Tuning Converts Weak Language Models to Strong Language Models. ICML 2024.
> > > > [5] Wu et al. (2024): Self-Play Preference Optimization for Language Model Alignment. CoRR abs/2405.00675.
> > > > [6] Gorbatovski et al. (2024): Learn your reference model for real good alignment. arXiv preprint arXiv:2404.09656.

---

> ### Author Response · Authors · 2024-12-04
> **We are grateful for the insightful discussions and the opportunity to refine our work through this constructive feedback process.**
>
> > ### 2. **The significance of theoretical guarantees**
>
> In our response [A11](https://openreview.net/forum?id=QqziJAdev9&noteId=596X5pzGxt), we emphasized that importance sampling is a widely adopted strategy in the literature on preference optimization and reinforcement learning.
>
> In our response [A12](https://openreview.net/forum?id=QqziJAdev9&noteId=MXDSpMJe8I), we analyzed the advantages of $\alpha$-DPO from a gradient perspective, highlighting its alignment with the optimization direction of online SimPO.
>
> Finally, we would like to emphasize that SimPO has already demonstrated its strengths in various experiments. Our method extends SimPO to personalized scenarios, with the parameter $\alpha$ serving as a critical control factor. Specifically:
> - The choice of $\alpha$ strikes a balance between two competing objectives. When $\alpha = 0$, the method simplifies to SimPO, where all training samples share the same target margin.
> - As $\alpha$ increases, the model places greater weight on samples with accurate reward estimates.
> - However, excessively large values of $\alpha$ cause the model to focus solely on a few samples with the most accurate reward estimates, neglecting the contributions of other samples, which can hinder the training process.
>
> This trade-off underscores the necessity of selecting an appropriate $\alpha$ to balance conservativeness and aggressiveness effectively.
>
>
> > ### 3. **The improvements are not significant.**
>
> In our response [A14](https://openreview.net/forum?id=QqziJAdev9&noteId=MXDSpMJe8I), we first highlighted that AlpacaEval2 and Arena-Hard are two of the most widely used benchmarks. These benchmarks have been validated using GPT-4 Turbo, achieving over 98% agreement with human evaluations. Therefore, we believe the scores derived from these benchmarks are sufficiently reliable.
>
> Furthermore, considering the development trends of current methodologies—from DPO to various approaches such as IPO, KTO, CPO, and the state-of-the-art SimPO introduced this year—our improvements over SimPO are both substantial and stable. We encourage the reviewers to re-examine the progress in existing techniques and recognize the significance of our contributions within this context.
>
> ---
>
> Your constructive feedback has been invaluable, and we are committed to leveraging this discussion to improve our work. Thank you again for the opportunity to address your questions and for considering our responses.

---

### Official Review · Reviewer_Jeuw · 2024-11-04

**Soundness:** 2
**Presentation:** 2
**Contribution:** 2
**Rating:** 5
**Confidence:** 4

**Summary:**

This paper finds that SimPO shares the same offset across all samples, which leads to suboptimal performance. To address this limitation, the authors construct a novel reference model based on SimPO and DPO, resulting in a novel method
$
\alpha\text{-DPO}.
$
The authors provide theoretical analysis on the lower bound and its connections to TDPO. Extensive experiments reveal the effectiveness of the proposed methods over various baselines and across different LLM structures.

**Strengths:**

1. Extensive experiments reveal that the $\alpha$-DPO outperforms various baselines across different LLM structures.

2. The authors try to connect the proposed methods with other existing alignment methods, which is interesting.

**Weaknesses:**

1. My major concerns is that the motivation of the proposed reference policy Eq.(8)  is not so clear. I understand that the authors want to construct a new reference policy that takes the advantages of DPO and SimPO. However, I think the authors should discuss more details why it is this form and how is its advantage. I understand that the authors provide some theoretical analysis to convince the readers. But I also think the motivation is also important.



2. The proofs should be discussed in more details. For example, in Lemma 4.2, in line 862, the authors use the Taylor expansion w.r.t
$
\log (A-\sigma B)
$
on
$
\alpha = 0.
$
This makes the lower bound only establishes around zero, which limits the theoretical contribution. Moreover, if
$
\alpha = 0,
$
then according to (8), we have
$
\hat{\pi}_{ref}(y \mid x)=U(y \mid x).
$
But as discussed lines 152-153, this is the reference function in SimPO? Then, does
$
\alpha
$-DPO degenerates to SimPO? If so, I think the Lemma 4.2 does not provide sufficient lower bound details over
$
\alpha
$-DPO.

Moreover, in lines 913 and 914, the authors state that "under the assumption that the reference policy
$
\pi_{ref}
$
has large errors", but it seems that I do not find it in the lemma, the authors mention it directly on the lemma and discuss why this assumption is mild.

**Questions:**

Please answer my questions mentioned above.

---

> ### Author Response · Authors · 2024-11-21
>
> We thank the reviewer for their thoughtful feedback and constructive suggestions. Below, we address your comments in detail.
>
> ---
> **Q1: My major concern is that the motivation of the proposed reference policy Eq.(8) is not so clear.**
>
> A1: The motivation for the proposed reference policy $\hat{\pi}\_{\text{ref}}(y|x)$ can be clarified as follows:
> 1. **Utility Theory Perspective:**
> The proposed $\hat{\pi}\_{\text{ref}}(y|x)$ is designed with the uniform distribution $U(y|x)$ as a baseline. The term $\left(\frac{\pi\_\theta(y|x)}{\pi\_{\text{ref}}(y|x)}\right)^\alpha$ dynamically adjusts the reward margin by balancing contributions from the policy and reference models. This mechanism can be interpreted through the lens of utility theory as relative attractiveness, enabling adaptive instance-specific reward modeling.
> 2. **Gradient Perspective:**
> By introducing $\hat{\pi}\_{\text{ref}}(y|x)$, the framework mitigates the label flipping issues found in DPO or SimPO.
> In the SimPO framework, the gradient is expressed as:
> $$\nabla\_\theta\mathcal{L}\_{\mathrm{SimPO}}(\pi\_\theta)=-\beta\mathbb{E}\_{(x,y\_w,y\_l)\sim\mathcal{D}}\left[s\_\theta\left(\frac1{|y\_w|}\nabla\_\theta\log\pi\_\theta(y\_w|x)-\frac1{|y\_l|}\nabla\_\theta\log\pi\_\theta(y\_l|x)\right)\right],$$
>    where
> $s\_\theta=\sigma\left(\frac\beta{|y\_l|}\log\pi\_\theta(y\_l|x)-\frac\beta{|y\_w|}\log\pi\_\theta(y\_w|x)+\gamma\right)$.
>    This formulation may amplify weights when the reward estimate is incorrect. By contrast, under $\alpha$-DPO:
> $$s\_\theta=\sigma\left(\frac\beta{|y\_l|}\log\pi\_\theta(y\_l|x)-\frac\beta{|y\_w|}\log\pi\_\theta(y\_w|x)+\gamma + \alpha M(x,y\_w,y\_l)\right),$$
>    the additional $\alpha M(x,y\_w,y\_l)$ component increases weight when the reward estimate is accurate, ensuring a more robust reward signal.
>
> 3. **Motivational Core:**
> The central goal of the proposed $\alpha$-DPO is to address the unreliability of the reference policy, as outlined in Section 3.1. By integrating the policy model into the reference model design, the quality of the reference model is enhanced, improving fine-tuning performance. Similar concepts have been explored in recent works [1][2]:
>
> `We have also provided a theoretical justification through reward difference estimation. For more details, please refer to our response (A4) to Reviewer 8xwP.`
>
> [1] Gorbatovski et al. (2024): Learn your reference model for real good alignment. arXiv preprint arXiv:2404.09656.
> [2] Liu et al. (2024): Understanding Reference Policies in Direct Preference Optimization. arXiv preprint arXiv:2407.13709.
>
> **Q2: The authors use the Taylor expansion w.r.t (\log(A-\alpha B)) on (\alpha=0). This makes the lower bound only established around zero, which limits the theoretical contribution.**
>
> A2: Thank you for pointing this out. We apologize for the imprecise explanation in the original submission. The correct statement is:
>
> **By expanding $\log\sigma(A-\alpha B)$ around $A$ using the first-order Taylor expansion, we obtain: ...**
>
> - The expansion is expanded around $A$, not $\alpha=0$.
> - Using the first-order Taylor expansion for $\log\sigma(A-\alpha B)$ at $A$, we derive:
>
> $$\log\sigma(A-\alpha B)\approx \log\sigma(A)+(\log\sigma(A))^{'}(-\alpha B)=\log\sigma(A)-\alpha B (1-\sigma(A)),$$
>
>   where the derivative of $\log\sigma(x)$, $(\log\sigma(x))'=1-\sigma(x)$, is used. This derivation ensures mathematical validity and is not limited to the neighborhood of $\alpha=0$. We have update the manuscript to reflect this clarification.
>
> **Q3: Moreover, in lines 913 and 914, the authors state that "under the assumption that the reference policy has large errors", but it seems that I do not find it in the lemma. The authors should mention it directly in the lemma and discuss why this assumption is mild.**
>
> A3: Thank you for pointing this out. The assumption that the reference policy has large errors serves as the motivation for proposing $\alpha$-DPO. This assumption is discussed in Section 3.1, where we highlight the unreliability of the reference policy in DPO. To make this point clearer, we have revised the manuscript to include this assumption explicitly in the lemma and expanded the discussion to emphasize its practical relevance and mildness. Specifically:
> - The unreliability of the reference policy arises due to the static nature of its design, which may not generalize well across diverse datasets or instances.
> - Incorporating the policy model into the reference policy design mitigates these limitations and improves the alignment between the reward model and policy fine-tuning objectives.
>
> This revision will ensure that the lemma is self-contained and directly addresses the context of the assumption.
>
> ---
> We hope these additional clarifications address your concerns comprehensively. Thank you again for your thoughtful review and the opportunity to improve our work.

---

> > ### Comment · Reviewer_Jeuw · 2024-12-03
> >
> > Thank you to the authors for the detailed responses. However, I still have some concerns:
> >
> >
> > 1. I understand that when the reward estimate is accurate, $\alpha M\left(x, y_w, y_l\right)$ increases the weight, ensuring a more robust reward signal. However, when the reward estimate is inaccurate, could this hinder training? Is there any empirical evidence demonstrating that the reward estimate is consistently accurate (or accurate most of the time)?
> >
> > 2. I understand that $\alpha$ can be non-zero, but lines 891 and 841 suggest that $\alpha$ still needs to be very small. This makes $\alpha$-DPO still need to be quite similar to SimPO to make the theory establish.

---

> ### Author Response · Authors · 2024-12-03
>
> Thank you for your comments. Below are our responses to your concerns:
>
> > **Q4: However, when the reward estimate is inaccurate, could this hinder training? Is there any empirical evidence demonstrating that the reward estimate is consistently accurate (or accurate most of the time)?**
>
> **A4:** When the reward estimate is inaccurate, it does not hinder training. Specifically, when the reward signal is incorrect (e.g., $ M(x, y_w, y_l) < 0 $), the weight decreases as defined by
> $$
> s_\theta = \sigma\left(\frac{\beta}{|y_l|}\log\pi_\theta(y_l|x) - \frac{\beta}{|y_w|}\log\pi_\theta(y_w|x) + \gamma + \alpha M(x, y_w, y_l)\right).
> $$
> This reduces the influence of incorrect signals, aligning with the design motivation of $\alpha$-DPO.
>
> Although we could not provide additional experiments within the rebuttal period due to time constraints, we have strong reasons to believe that the reward estimate is consistently accurate. Specifically, the quality of the reference model improves with the fine-tuned $\pi_\theta$, as shown in our theoretical and empirical analyses.
>
>
> > **Q5: I understand that $\alpha$ can be non-zero, but lines 891 and 841 suggest that $\alpha$ still needs to be very small. This makes $\alpha$-DPO still need to be quite similar to SimPO to make the theory establish.**
>
>
> **A5:** Based on both theoretical considerations (to maintain similarity to SimPO and support the theoretical framework) and empirical results (validated by the parameter selections in Appendix Table 3 and Table 4), it appears that $\alpha$ is effective when kept small. Our findings suggest that using a small $\alpha$, such as the default value of $1\text{e-}2$, provides stable and significant performance improvements, aligning well with theoretical expectations and experimental observations.
>
> The choice of $\alpha$ reflects a trade-off. When $\alpha$ is set to 0, the approach reduces to SimPO. As $\alpha$ increases, the model places greater emphasis on samples where the reward estimate is correct. However, if $\alpha$ becomes too large, the model may become overly conservative, which can negatively affect training by limiting its ability to explore diverse solutions. This trade-off underscores the importance of selecting an appropriate $\alpha$ to balance alignment and adaptability.

---

> > ### Author Response · Authors · 2024-12-04
> > **We are grateful for the insightful discussions and the opportunity to refine our work through this constructive feedback process.**
> >
> > We greatly appreciate the opportunity to engage in a substantive dialogue with you, aimed at further improving the quality and clarity of our work. Below, we provide a detailed summary addressing your key concerns. We hope that this explanation will sufficiently address your questions and merit your endorsement.
> >
> > > ### 1. **Motivation for the proposed reference policy (Eq. 8)**
> >
> > First, let us clarify that the primary motivation for proposing the reference policy in Eq. (8) arises from the limitations of the conventional $\pi\_{\text{ref}} = \pi\_{\text{SFT}}$ setup, which often exhibits significant randomness and errors. Addressing this limitation is a core objective of our work, a perspective that is similarly reflected in SimPO [1] (Section 3.2 and Figure 4b) and Liu et al. [2] (Section 5).
> >
> > Our goal is thus to design a more robust and effective reference policy, as expressed in Eq. (8). To provide additional context, we elaborate on the following key points:
> >
> > - **Alignment with online SimPO optimization.** Preference optimization with online elements (e.g., online AI feedback [3] and self-play [4][5]) is widely acknowledged to improve model alignment by generating new data during training. However, the high computational cost and instability of purely online methods limit their practicality in industrial applications. Our work explores whether online-like behavior can be emulated in a classical offline setting. Theoretical analysis based on importance sampling is provided to bridge this gap.
> >
> > - **Structural advantages of $r(x, y\_w) - r(x, y\_l) - \delta$.** A major contribution of token-level DPO is the introduction of the form $r(x, y\_w) - r(x, y\_l) - \delta$, which is similar to DPO with an offset and helps control the KL divergence. Our $\alpha$-DPO framework leverages the form $r(x, y\_w) - r(x, y\_l) - M$, which enhances performance by using $M$ instead of $\delta$. Appendix Table 6 provides empirical evidence, showing the performance advantages of $\alpha$-DPO over TDPO.
> >
> > - **Mitigating the impact of label flipping noise.** Within the SimPO framework, the gradient is expressed as:
> >   $$
> >   \nabla\_\theta \mathcal{L}\_{\mathrm{SimPO}}(\pi\_\theta) = -\beta \mathbb{E}\_{(x, y\_w, y\_l) \sim \mathcal{D}} \left[ s\_\theta \left( \frac{1}{|y\_w|} \nabla\_\theta \log \pi\_\theta(y\_w|x) - \frac{1}{|y\_l|} \nabla\_\theta \log \pi\_\theta(y\_l|x) \right) \right],
> >   $$
> >   where
> >   $$
> >   s\_\theta = \sigma \left( \frac{\beta}{|y\_l|} \log \pi\_\theta(y\_l|x) - \frac{\beta}{|y\_w|} \log \pi\_\theta(y\_w|x) + \gamma \right).
> >   $$
> >   This formulation amplifies weights when the reward estimate is inaccurate (e.g., $\log \pi\_\theta(y\_l|x) - \log \pi\_\theta(y\_w|x) > 0$), increasing noise in gradients. In contrast, $\alpha$-DPO introduces an additional term:
> >   $$
> >   s\_\theta = \sigma \left( \frac{\beta}{|y\_l|} \log \pi\_\theta(y\_l|x) - \frac{\beta}{|y\_w|} \log \pi\_\theta(y\_w|x) + \gamma + \alpha M(x, y\_w, y\_l) \right).
> >   $$
> >   This term increases the weight when the reward estimate is accurate and decreases it when the reward estimate is inaccurate, mitigating noise amplification.
> >
> > - **Improving reference policy quality.** This work directly addresses the unreliability of reference policies (Section 3.1). By integrating the policy model into the reference model's design, we enhance the quality of the reference model and improve fine-tuning performance. Similar ideas have been explored in recent works [2][6], highlighting the broader relevance of this approach.
> >
> > References:
> > [1] Meng et al. (2024): Simpo: Simple preference optimization with a reference-free reward. NeurIPS 2024.
> > [2] Liu et al. (2024): Understanding Reference Policies in Direct Preference Optimization. arXiv preprint arXiv:2407.13709.
> > [3] Guo et al. (2024): Direct Language Model Alignment from Online AI Feedback. CoRR abs/2402.04792.
> > [4] Chen et al. (2024): Self-Play Fine-Tuning Converts Weak Language Models to Strong Language Models. ICML 2024.
> > [5] Wu et al. (2024): Self-Play Preference Optimization for Language Model Alignment. CoRR abs/2405.00675.
> > [6] Gorbatovski et al. (2024): Learn your reference model for real good alignment. arXiv preprint arXiv:2404.09656.

---

> > ### Author Response · Authors · 2024-12-04
> > **We are grateful for the insightful discussions and the opportunity to refine our work through this constructive feedback process.**
> >
> > > ### 2. **How does $\alpha M(x, y\_w, y\_l)$ affect the training process?**
> >
> > As elaborated in [A4](https://openreview.net/forum?id=QqziJAdev9&noteId=g7xaCtWbwc), we provide an intuitive understanding of how $\alpha M(x, y\_w, y\_l)$ aids training. When $M(x, y\_w, y\_l) > 0$ (reward signal is correct), the model assigns a higher weight to the sample for updating. Conversely, when $M(x, y\_w, y\_l) < 0$ (reward signal is incorrect), the model reduces the weight, mitigating the impact of erroneous signals.
> >
> >
> > > ### 3. **Why does $\alpha$ need to be very small?**
> >
> > **The $\alpha$ is not necessarily better when smaller; rather, it represents a trade-off**. Specifically:
> >
> > The choice of $\alpha$ balances two competing objectives. When $\alpha = 0$, the method simplifies to SimPO, where all training samples are assigned the same target margin. As $\alpha$ increases, the model places greater emphasis on samples with accurate reward estimates. However, excessively large values of $\alpha$ cause the model to focus solely on a few samples with the most accurate reward estimates, neglecting the contributions of other samples, which can hinder the training process. This trade-off highlights the critical need to select an appropriate $\alpha$ to balance conservativeness and aggressiveness.
> >
> > Finally, as shown in Figure 4(a) of our experiments, model performance initially improves and then declines as $\alpha$ increases, aligning well with the theoretical understanding of this trade-off.
> >
> > ---
> >
> > Your constructive feedback has been invaluable, and we are committed to leveraging this discussion to improve our work. Thank you again for the opportunity to address your questions and for considering our responses.

---

### Official Review · Reviewer_5mgV · 2024-11-10

**Soundness:** 2
**Presentation:** 2
**Contribution:** 2
**Rating:** 6
**Confidence:** 4

**Summary:**

The paper proposes a new loss for fine-tuning language models based on preference data in a reward-free fashion along the lines of DPO and SimPO. In particular, the paper aims to address the limitations of the DPO and SimPO loss by introducing an instant dependent offset in the SimPO loss. The loss and relations to SimPO and token-level DPO loss have been studied. Experiments have been conducted using standard benchmarks and ablations with various hyperparameters.

**Strengths:**

Strength:
1.	Considering an instant-specific margin compared to SimPO is interesting.
2.	Experimental Setup, analysis along ablations is exhaustive
3.	Reward differences between chosen and rejected responses and Log probabilities of the chosen response w.r.t. the fine-tuned model is shown. Further, KL divergence from the base model has been studied.
4.	Limitations of DPO and SimPO are -discussed.

I have read the rebuttal by the authors and thank them for clarification. Those helped me better understand and as such I raised my score.

**Weaknesses:**

1.	To the best of my understanding, Theorem 3.1 seems not correct? In particular, $\gamma$ has been defined to be log-difference of the chosen vs rejected response w.r.t. uniform policy. However, as uniform policy assigns equal probabilities to all responses, this directly implies $\gamma=0$. As the entire Theorem 3.1 depends on this to connect SimPO to a uniform reference distribution, this theorem seems meaningless. Can the authors please elaborate and explain this point? Maybe I misunderstood something?
2.	In their derivation of $\alpha-$DPO objective, they again use $\gamma$ to be log-difference of the chosen vs rejected response w.r.t. uniform policy and then claim that when $\alpha$ is 0, they get back the SimPO loss. That is again confusing, as detailed above. Can you kindly elaborate?
3.	Their Margin term $M$ is the same as the inner term in DPO loss. So essentially, their proposed loss in Eqn.12 combines SimPO and DPO loss with an extra stop gradient on the DPO loss. Is this correct?
4.	In Section 4.1, they aim to connect $\alpha-$DPO with SimPO loss. However, they introduce an importance sampling correction term for the online SimPO loss. It is unclear how that is relevant to Lemma 4.2 or any other aspect of the paper. Can you please help me better understand this part?
5. I am not sure how Lemma 4.3 contributes. The margin term M is same as inner term of DPO. Hence, it is evident that is close to $\delta$ from token-level DPO loss. Further, their claim that they improve upon token-level DPO is not correct. Token-level DPO was proposed to improve upon DPO but with more computational cost. One can’t revert back to sequence-level loss as in DPO from token-level loss and claim improvement/novelty. All the previous works use this sequence-level loss.
6. It feels that several parts of the paper, including discussions, seem to be written using AI tools, e.g., a discussion of Theorem 3.1 and Lemma 4.1. I could be very well wrong, just trying to understand

**Questions:**

Please see above

---

> ### Author Response · Authors · 2024-11-21
>
> We thank the reviewer for their thoughtful feedback and constructive suggestions. Below, we address your comments in detail.
>
> ---
>
> **Q1: However, as uniform policy assigns equal probabilities to all responses, this directly implies $\gamma=0$.**
>
> A1: **We first want to jusitfy that the uniform distribution assumption is proper, and the uniform distribution does not lead to $\gamma = 0$.** The uniform distribution assumption on a LLM describes this model as not having any alignment w.r.t human preference. It is commonly happening in the LLM without enough high equality instruction tuning.
> The positive/negative samples are generated via reject sampling procedures on LLM and the corresponding distributions are different.
>
> Let's revisit the acquisition method for these samples (e.g., using princeton-nlp/llama3-ultrafeedback dataset):
> For each prompt $x$, we generate 5 responses using the SFT model with a sampling temperature of 0.8. We then score these responses with llm-blender/PairRM and select the highest-scoring one as $y\_w$ and the lowest-scoring one as $y\_l$.
>
> For the highest-scoring sample $y\_w$, it follows a new conditional distribution:
>
> $$\pi(y \mid x, \text{score}(y) = \max(\{\text{score}(y\_1), \ldots, \text{score}(y\_5)\}))\approx\pi(win|x),$$
>
> where the last part uses the assumption that for a given x the generative model has a uniform chance to generate all possible y.
> Similarly, for the lowest-scoring sample $y\_l$, it follows:
>
> $$\pi(y \mid x, \text{score}(y) = \min(\{\text{score}(y\_1), \ldots, \text{score}(y\_5)\}))\approx \pi(lose|x)$$
>
> Each response is generated from the same conditional distribution $\pi\_{\text{SFT}}$, implying the same distribution during generation. **However, by scoring and selecting using llm-blender/PairRM, there is an inherent selection bias.** In the above setting, the $\gamma$ value is
> $\gamma \approx \log \pi (win|x) - \log \pi(lose|x)$.
>
>
> By further assuming this situation uniformly happens over all x, we reduce to the SimPO configuration.
> Thus, the statement "uniform policy assigns equal probabilities to all responses" is inaccurate.
>
>
>
> **Q2: They claim that when it is 0, they get back the SimPO loss.**
>
> A2: Please note that according to the discussion in the answer to Q1, the uniform assumption doesn't lead to $\gamma = 0$. Based on the definition in Eq (8), when $\alpha = 0$, $\hat{\pi}\_{\textrm{ref}}$ becomes the uniform distribution. Similarly to A1, the uniform distribution can mitigate the unreliability of $\pi\_{\text{ref}}$, but it still lacks sample personalization and remains suboptimal. Our $\alpha$-DPO objective is designed to incorporate the strengths of both DPO and SimPO. We thank the reviewer for pointing it out and will add the discussion in the Section 3.1 to further clarify it.
>
> **Q3: Their proposed loss in Eqn.12 combines SimPO and DPO loss with an extra stop gradient on the DPO loss.**
>
> A3: Indeed, from a formal perspective, the $\alpha$-DPO appears as DPO with an additional stop gradient on the Margin term. However, this result demonstrates that SimPO with an offset ($r(x,y\_w)-r(x,y\_l)-M$) can further enhance performance, presenting theoretical benefits (c.f. Section 4) and relationships that, although concise, are effective.
>
> ``We have also provided a theoretical justification through reward difference estimation. For more details, please refer to our response (A4) to Reviewer 8xwP.``

---

> ### Author Response · Authors · 2024-11-21
>
> **Q4: They introduce an importance sampling correction term for the online SimPO loss. How is that relevant to Lemma 4.2 or any other part of the paper?**
>
> A4: It is well known that preference optimization with online ingredients, particularly methods like online AI feedback (OAIF) [1] and self-play [2,3], enhances model alignment by generating new data during training.
> However, due to the computation cost sample regeneration and training stability issues, the pure online type preference optimization methods are not well applied in industrial practice.  Based on this observation, we thus proposed to study an interesting research topic, i.e., `can we mimic the online feature in the classic offline setting?` Our theoretical analysis is trying to close the gap with the importance sampling trick.
>
> In particular, the underlying logic here first defines the expression for online SimPO, which is characterized by the continuous updating of sampled data during training. In this process, the data transitions from the offline set $(y\_w, y\_l) \in D$ to the online set $(y\_w, y\_l) \in \pi\_\theta$. This operation can be viewed as an important sampling method. Interestingly, this key technique of importance sampling aligns with the optimization direction of $\alpha$-DPO loss, with both approaches converging in the scenario where $\alpha \to 0$. Consequently, Lemma 4.2 aims to establish a connection between $\alpha$-DPO and the Online SimPO loss, thereby facilitating the integration of these concepts.
>
> [1] Guo et al. Direct Language Model Alignment from Online AI Feedback. CoRR abs/2402.04792 (2024)
>
> [2] Chen et al. Self-Play Fine-Tuning Converts Weak Language Models to Strong Language Models. ICML 2024.
>
> [3] Wu et al. Self-Play Preference Optimization for Language Model Alignment. CoRR abs/2405.00675 (2024)
>
> **Q5: I am not sure how Lemma 4.3 contributes. Further, their claim that they improve upon token-level DPO is not correct.**
>
> A5: There seems to be a misconception that needs clarification. The core contribution of token-level DPO is the introduction of the form $r(x,y\_w)-r(x,y\_l)-\delta$, similar to DPO with an offset, enabling control over KL divergence. The $\alpha$-DPO follows the form SimPO with an offset $r(x,y\_w)-r(x,y\_l)-M$, providing performance enhancement and showing that $M$ is more effective than $\delta$. Appendix Table 6 supports this with performance comparisons between TDPO and $\alpha$-DPO.
> These findings illustrate:
> 1. Adding an offset to DPO and its variants is a successful strategy, applicable to both token-level DPO and $\alpha$-DPO.
> 2. The choice of offset is still undecided. Under the premise of an unreliable reference model in the SimPO concept, $M$ outperforms $\delta$, offering valuable insights.
>
> ---
>
> We hope these additional clarifications address your concerns comprehensively. Thank you again for your thoughtful review and the opportunity to improve our work.

---

> ### Author Response · Authors · 2024-12-04
>
> We sincerely thank you for your support and for raising the score of our work. We deeply appreciate your recognition of our contributions to the $\alpha$-DPO. Regarding your valuable suggestions on improving the presentation, we will incorporate them into the final version. Thank you again for your thoughtful feedback, which has been instrumental in enhancing the quality of our work.

---

### Author Response · Authors · 2024-12-04

We thank all reviewers for their valuable and insightful feedback.

We are encouraged that the reviewers found our paper meaningful (Reviewers $\color{red}{\text{5mgV}}$, $\color{green}{\text{Jeuw}}$, $\color{black}{\text{EUrj}}$, $\color{orange}{\text{8xwP}}$). Furthermore, we are grateful that the reviewers recognized the simplicity and effectiveness of our proposed $\alpha$-DPO algorithm (Reviewers $\color{red}{\text{5mgV}}$, $\color{green}{\text{Jeuw}}$, $\color{blue}{\text{RjJZ}}$, $\color{black}{\text{EUrj}}$, $\color{orange}{\text{8xwP}}$). We also appreciate that several reviewers found our paper well-written and easy to follow (Reviewers $\color{blue}{\text{RjJZ}}$, $\color{black}{\text{EUrj}}$, $\color{orange}{\text{8xwP}}$).

We acknowledge the reviewers' constructive comments and critiques, which helped us to identify areas for improvement. Below, we summarize our discussions and detailed responses to each reviewer’s feedback:

- **Reviewer $\color{red}{\text{5mgV}}$**: We provided a detailed explanation of the uniform policy and clarified why it does not result in $\gamma=0$. Considering your positive rating, we believe that your concerns have been addressed. Thank you for your encouraging feedback.

- **Reviewer $\color{green}{\text{Jeuw}}$**: We elaborated on the motivation behind the proposed reference policy in Eq. (8) from several perspectives, including its connection to online algorithms, the advantage of DPO with an offset, and its ability to mitigate label-flipped noise. Thank you for your constructive and valuable comments.

- **Reviewer $\color{blue}{\text{RjJZ}}$**: We extended the explanation of the proposed reference policy in Eq. (8) by highlighting its connections to online algorithms, the offset advantages of DPO, and its robustness to label-flipped noise. Additionally, we included experimental results on benchmarks such as MT-Bench, MMLU, GSM8K, and TruthfulQA. We believe these extensive comparisons provide strong evidence for the reliability of our method. Thank you for your constructive feedback.

- **Reviewer $\color{black}{\text{EUrj}}$**: We clarified the performance trends of different metrics under varying $\alpha$ values, demonstrating that $\alpha$-DPO achieves superior performance across all benchmarks with a fixed $\alpha$ value. Considering your positive rating, we believe your concerns have been adequately addressed. Thank you for your encouraging feedback.

- **Reviewer $\color{orange}{\text{8xwP}}$**: We provided an in-depth explanation of the motivation behind the proposed reference policy in Eq. (8), focusing on its connection to online algorithms, the advantages of DPO with an offset, and its capability to mitigate label-flipped noise. Thank you for your constructive comments.

As the author-reviewer discussion phase comes to a close, we sincerely hope that our responses and improvements have addressed your concerns effectively. If there are any remaining questions, we are more than happy to provide further clarifications. Once again, we thank all reviewers for their thoughtful efforts in improving the quality of our work.

---

### Meta-Review · Area_Chair_7ey1 · 2024-12-20

**Metareview:**

This paper proposes a novel strategy for LLM alignment designed to address the limitations of simple preference optimization (SPO) and direct policy optimization (DPO). There were remaining concerns about the clarity of the exposition, the scope of the theoretical contributions, as well as the significance of the empirical results. Thus, the paper still requires significant revision before it can be considered for acceptance.

**Additional Comments On Reviewer Discussion:**

There was significant discussion during the rebuttal period. During this time, the authors addressed some of the technical concerns regarding the correctness of the paper. However, there were lingering concerns about the scope of the contribution, especially from Reviewer RjJZ but also to a lesser degree from the other reviewers.

---

### Decision · Program_Chairs · 2025-01-22

Reject